# Graph-Guided Network for Irregularly Sampled Multivariate Time Series

**Xiang Zhang**
Harvard University
`xiang_zhang@hms.harvard.edu`

**Marko Zeman**
University of Ljubljana
`marko.zeman@fri.uni-lj.si`

**Theodoros Tsiligkaridis**
MIT Lincoln Laboratory
`ttsili@ll.mit.edu`

**Marinka Zitnik**
Harvard University
`marinka@hms.harvard.edu`

## Abstract

In many domains, including healthcare, biology, and climate science, time series are irregularly sampled with varying time intervals between successive readouts and different subsets of variables (sensors) observed at different time points. Here, we introduce RAINDROP, a graph neural network that embeds irregularly sampled and multivariate time series while also learning the dynamics of sensors purely from observational data. RAINDROP represents every sample as a separate sensor graph and models time-varying dependencies between sensors with a novel message passing operator. It estimates the latent sensor graph structure and leverages the structure together with nearby observations to predict misaligned readouts. This model can be interpreted as a graph neural network that sends messages over graphs that are optimized for capturing time-varying dependencies among sensors. We use RAINDROP to classify time series and interpret temporal dynamics on three healthcare and human activity datasets. RAINDROP outperforms state-of-the-art methods by up to 11.4% (absolute F1-score points), including techniques that deal with irregular sampling using fixed discretization and set functions. RAINDROP shows superiority in diverse setups, including challenging leave-sensor-out settings.

## 1 Introduction

Multivariate time series are prevalent in a variety of domains, including healthcare, space science, cyber security, biology, and finance (Ravuri et al., 2021; Sousa et al., 2020; Sezer et al., 2020; Fawaz et al., 2019). Practical issues often exist in collecting sensor measurements that lead to various types of irregularities caused by missing observations, such as saving costs, sensor failures, external forces in physical systems, medical interventions, to name a few (Choi et al., 2020). While temporal machine learning models typically assume fully observed and fixed-size inputs, irregularly sampled time series raise considerable challenges (Shukla & Marlin, 2021; Hu et al., 2021). For example, observations of different sensors might not be aligned, time intervals among adjacent observations are different across sensors, and different samples have different numbers of observations for different subsets of sensors recorded at different time points (Horn et al., 2020; Wang et al., 2011).

Prior methods for dealing with irregularly sampled time series involve filling in missing values using interpolation, kernel methods, and probabilistic approaches (Schafer & Graham, 2002). However, the absence of observations can be informative on its own (Little & Rubin, 2014) and thus imputing missing observations is not necessarily beneficial (Agniel et al., 2018). While modern techniques involve recurrent neural network architectures (*e.g.*, RNN, LSTM, GRU) (Cho et al., 2014) and transformers (Vaswani et al., 2017), they are restricted to regular sampling or assume aligned measurements across modalities. For misaligned measurements, existing methods tend to rely on a two-stage approach that first imputes missing values to produce a regularly-sampled dataset and then optimizes a model of choice for downstream performance. This decoupled approach does not fully exploit informative missingness patterns or deal with irregular sampling, thus producing suboptimal

performance (Wells et al., 2013; Li & Marlin, 2016). Thus, recent methods circumvent the imputation stage and directly model irregularly sampled time series (Che et al., 2018; Horn et al., 2020).

Previous studies (Wu et al., 2021; Li et al., 2020a; Zhang et al., 2019) have noted that inter-sensor correlations bring rich information in modeling time series. However, only few studies consider relational structure of irregularly sampled time series, and those which do have limited ability in capturing inter-sensor connections (Wu et al., 2021; Shukla & Marlin, 2018). In contrast, we integrate recent advances in graph neural networks to take advantage of relational structure among sensors. We learn latent graphs from multivariate time series and model time-varying inter-sensor dependencies through neural message passing, establishing graph neural networks as a way to model sample-varying and time-varying structure in complex time series.

**Present work.** To address the characteristics of irregularly sampled time series, we propose to model temporal dynamics of sensor dependencies and how those relationships evolve over time. Our intuitive assumption is that the observed sensors can indicate how the unobserved sensors currently behave, which can further improve the representation learning of irregular multivariate time series. We develop RAINDROP[1], a graph neural network that leverages relational structure to embed and classify irregularly sampled multivariate time series. RAINDROP takes *samples* as input, each sample containing multiple *sensors* and each sensor consisting of irregularly recorded *observations* (*e.g.*, in clinical data, an individual patient's state of health is recorded at irregular time intervals with different subsets of sensors observed at different times). RAINDROP model is inspired by how raindrops hit a surface at varying times and create ripple effects that propagate through the surface. Mathematically, in RAINDROP, observations (*i.e.*, raindrops) hit a sensor graph (*i.e.*, surface) asynchronously

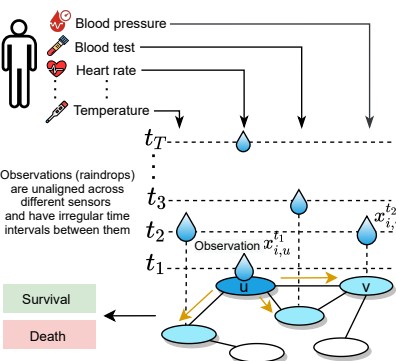

**Figure 1:** The RAINDROP approach. For sample $\mathcal{S}_i$, sensor $u$ is recorded at time $t_1$ as value $x_{i,u}^{t_1}$, triggering a propagation and transformation of neural messages along edges of $\mathcal{S}_i$'s sensor dependency graph.

and at irregular time intervals. Every observation is processed by passing messages to neighboring sensors (*i.e.*, creating ripples), taking into account the learned sensor dependencies (Figure 1). As such, RAINDROP can handle misaligned observations, varying time gaps, arbitrary numbers of observations, and produce multi-scale embeddings via a novel hierarchical attention.

We represent dependencies with a separate sensor graph for every sample wherein nodes indicate sensors and edges denote relationships between them. Sensor graphs are latent in the sense that graph connectivity is learned by RAINDROP purely from observational time series. In addition to capturing sensor dependencies within each sample, RAINDROP i) takes advantage of similarities between different samples by sharing parameters when calculating attention weights, and ii) considers importance of sequential sensor observations via temporal attention.

RAINDROP adaptively estimates observations based on both neighboring readouts in the temporal domain and similar sensors as determined by the connectivity of optimized sensor graphs. We compare RAINDROP to five state-of-the-art methods on two healthcare datasets and an activity recognition dataset across three experimental settings, including a setup where a subset of sensors in the test set is malfunctioning (*i.e.*, have no readouts at all). Experiments show that RAINDROP outperforms baselines on all datasets with an average AUROC improvement of 3.5% in absolute points on various classification tasks. Further, RAINDROP improves prior work by a 9.3% margin (absolute points in accuracy) when varying subsets of sensors malfunction.

## 2 RELATED WORK

Our work here builds on time-series representation learning and notions of graph neural networks and attempts to resolve them by developing a single, unified approach for analysis of complex time series.

**Learning with irregularly sampled multivariate time series.** Irregular time series are characterized by varying time intervals between adjacent observations (Zerveas et al., 2021; Tipirneni &

---

[1]Code and datasets are available at https://github.com/mims-harvard/Raindrop.

Reddy, 2021; Chen et al., 2020). In a multivariate case, irregularity means that observations can be misaligned across different sensors, which can further complicate the analysis. Further, because of a multitude of sampling frequencies and varying time intervals, the number of observations can also vary considerably across samples (Fang & Wang, 2020; Kidger et al., 2020). Predominant downstream tasks for time series are classification (*i.e.*, predicting a label for a given sample, *e.g.*, Tan et al. (2020); Ma et al. (2020)) and forecasting (*i.e.*, anticipating future observations based on historical observations, *e.g.*, Wu et al. (2020a)). The above mentioned characteristics create considerable challenges for models that expect well-aligned and fixed-size inputs (Shukla & Marlin, 2020). An intuitive way to deal with irregular time series is to impute missing values and process them as regular time series (Mikalsen et al., 2021; Li & Marlin, 2020; Shan & Oliva, 2021). However, imputation methods can distort the underlying distribution and lead to unwanted distribution shifts. To this end, recent methods directly learn from irregularly sampled time series (Chen et al., 2018). For example, Che et al. (2018) develop a decay mechanism based on gated recurrent units (*GRU-D*) and binary masking to capture long-range temporal dependencies. *SeFT* (Horn et al., 2020) takes a set-based approach and transforms irregularly sampled time series datasets into sets of observations modeled by set functions insensitive to misalignment. *mTAND* (Shukla & Marlin, 2021) leverages a multi-time attention mechanism to learn temporal similarity from non-uniformly collected measurements and produce continuous-time embeddings. *IP-Net* (Shukla & Marlin, 2018) and *DGM²* (Wu et al., 2021) adopt imputation to interpolate irregular time series against a set of reference points using a kernel-based approach. The learned inter-sensor relations are static ignoring sample-specific and time-specific characteristics. In contrast with the above methods, RAINDROP leverages dynamic graphs to address the characteristics of irregular time series and produce high-quality representations.

**Learning with graphs and neural message passing.** There has been a surge of interest in applying neural networks to graphs, leading to the development of graph embeddings (Zhou et al., 2020; Li et al., 2021), graph neural networks (Wu et al., 2020b), and message passing neural networks (Gilmer et al., 2017). To address the challenges of irregular time series, RAINDROP specifies a message passing strategy to exchange neural message along edges of sensor graphs and deal with misaligned sensor readouts (Riba et al., 2018; Nikolentzos et al., 2020; Galkin et al., 2020; Fey et al., 2020; Lin et al., 2018; Zhang et al., 2020). In particular, RAINDROP considers message passing on latent sensor graphs, each graph describing a different sample (*e.g.*, patient, Figure 1), and it specifies a message-passing network with learnable adjacency matrices. The key difference with the predominant use of message passing is that RAINDROP uses it to estimate edges (dependencies) between sensors rather than applying it on a fixed, apriori-given graph. To the best of our knowledge, prior work did not utilize sensor dependencies for irregularly sampled time series. While prior work used message passing for regular time series (Wang et al., 2020; Wu et al., 2020c; Kalinicheva et al., 2020; Zha et al., 2022), its utility for irregularly sampled time series has not yet been studied.

## 3 RAINDROP

Let $\mathcal{D} = \{(\mathcal{S}_i, y_i) \mid i = 1, \ldots, N\}$ denote an irregular time series dataset with $N$ labeled samples (Figure 2). Every sample $\mathcal{S}_i$ is an irregular multivariate time series with a corresponding label $y_i \in \{1, \ldots, C\}$, indicating which of the $C$ classes $\mathcal{S}_i$ is associated with. Each sample contains $M$ non-uniformly measured sensors that are denoted as $u$, $v$, etc. RAINDROP can also work on samples with only a subset of active sensors (see Sec. 4.1). Each sensor is given by a sequence of observations ordered by time. For sensor $u$ in sample $\mathcal{S}_i$, we denote a single observation as a tuple $(t, x_{i,u}^t)$, meaning that sensor $u$ was recorded with value $x_{i,u}^t \in \mathbb{R}$ at timestamp $t \in \mathbb{R}^+$. We omit sample index $i$ and sensor index $u$ in timestamp $t$. Sensor observations are irregularly recorded, meaning that time intervals between successive observations can vary across sensors. For sensor $u$ in sample $\mathcal{S}_i$, we use $\mathcal{T}_{i,u}$ to denote the set of timestamps that $u$, or at least one of $u$'s $L$-hop neighbors ($L$ is the number of layers in RAINDROP's message passing) is recorded. We use $||$ and $^T$ to denote concatenation and transpose, respectively. We omit layer index $l \in \{1, \ldots, L\}$ for simplicity when clear from the text.

**Problem (Representation learning for irregularly sampled multivariate time series).** *A dataset $\mathcal{D}$ of irregularly sampled multivariate time series is given, where each sample $\mathcal{S}_i$ has multiple sensors and each sensor has a variable number of observations. RAINDROP learns a function $f : \mathcal{S}_i \to \boldsymbol{z}_i$ that maps $\mathcal{S}_i$ to a fixed-length representation $\boldsymbol{z}_i$ suitable for downstream task of interest, such as classification. Using learned $\boldsymbol{z}_i$, RAINDROP can predict label $\hat{y}_i \in \{1, \ldots, C\}$ for $\mathcal{S}_i$.*

RAINDROP learns informative embeddings for irregularly samples time series. The learned embeddings capture temporal patterns of irregular observations and explicitly consider varying dependencies between sensors. While we focus on time-series classification in this work, the proposed method can be easily extended to broader applications such as regression, clustering and generation tasks.

## 3.1 OVERVIEW OF RAINDROP

RAINDROP aims to learn a fixed-dimensional embedding $z_i$ for a given sample $\mathcal{S}_i$ and predict the associated label $\hat{y}_i$. To this end, it generates sample embeddings using a hierarchical architecture composed of three levels to model observations (sensor readouts), sensors, and whole samples (Figure 2). Without loss of generality, we describe RAINDROP's procedure as if observations arrive one at a time (one sensor is observed at time $t$ and other sensors do not have observations). If there are multiple observations at the same time, RAINDROP can effortlessly process them in parallel.

RAINDROP first constructs a graph for every sample where nodes represent sensors and edges indicate relations between sensors (Sec. 3.2). We use $\mathcal{G}_i$ to denote the sensor graph for sample $\mathcal{S}_i$ and $e_{i,uv}$ to represent the weight of a directed edge from sensor $u$ to sensor $v$ in $\mathcal{G}_i$. Sensor graphs are automatically optimized considering sample-wise and time-wise specificity.

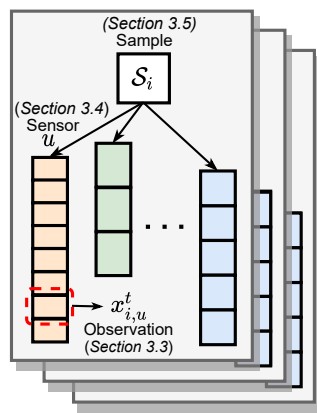

**Figure 2:** Hierarchical structure of irregular multivariate time series dataset. RAINDROP embeds individual observations considering inter-sensor dependencies (Sec. 3.3), aggregates them into a sensor embedding using temporal attention (Sec. 3.4), and finally integrates sensor embeddings into a sample embedding (Sec. 3.5).

The key idea of RAINDROP is to borrow information from $u$'s neighbors based on estimated relationships between $u$ and other sensors. This is achieved via message passing carried out on $\mathcal{S}_i$'s dependency graph and initiated at node $u$ in the graph. When an observation $(t, x_{i,u}^t)$ is recorded for sample $\mathcal{S}_i$ at time $t$, RAINDROP first embeds the observation at *active* sensor $u$ (*i.e.*, sensor whose value was recorded) and then propagates messages (*i.e.*, the observation embeddings) from $u$ to neighboring sensors along edges in sensor dependency graph $\mathcal{G}_i$. As a result, recording the value of $u$ can affect $u$'s embedding as well as embeddings of other sensors that related to $u$ (Sec. 3.3). Finally, RAINDROP generates sensor embeddings by aggregating all observation embeddings for each sensor (across all timestamps) using temporal attention weights (Sec. 3.4). At last, RAINDROP embeds sample $\mathcal{S}_i$ based on sensor embeddings (Sec. 3.5) and feeds the sample embedding into a downstream predictor.

## 3.2 CONSTRUCTING SENSOR DEPENDENCY GRAPHS

We build a directed weighted graph $\mathcal{G}_i = \{\mathcal{V}, \mathcal{E}_i\}$ for every sample $\mathcal{S}_i$ and refer to it as the *sensor dependency graph* for $\mathcal{S}_i$. Nodes $\mathcal{V}$ represent sensors and edges $\mathcal{E}_i$ describe dependencies between sensors in sample $\mathcal{S}_i$ that RAINDROP infers. As we show in experiments, RAINDROP can be directly used with samples that only contain a subset of sensors in $\mathcal{V}$. We denote edge from $u$ to $v$ as a triplet $(u, e_{i,uv}, v)$, where $e_{i,uv} \in [0, 1]$ represents the strength of relationship between sensors $u$ and $v$ in sample $\mathcal{S}_i$. Edge $(u, e_{i,uv}, v)$ describes the relationship between $u$ and $v$: when $u$ receives an observation, it will send a neural message to $v$ following edge $e_{i,uv}$. If $e_{i,uv} = 0$, there is no exchange of neural information between $u$ and $v$, indicating that the two sensors are unrelated. We assume that the importance of $u$ to $v$ is different than the importance of $v$ to $u$, and so we treat sensor dependency graphs as directed, *i.e.*, $e_{i,uv} \neq e_{i,vu}$. All graphs are initialized as fully-connected graphs (*i.e.*, $e_{i,uv} = 1$ for any $u$, $v$ and $\mathcal{S}_i$) and edge weights $e_{i,uv}$ are updated following Eq. 3 during model training. If available, it is easy to integrate additional domain knowledge into graph initialization.

## 3.3 GENERATING EMBEDDINGS OF INDIVIDUAL OBSERVATIONS

Let $u$ indicate active sensor at time $t \in \mathcal{T}_{i,u}$, i.e., sensor whose value $x_{i,u}^t$ is observed at $t$, and let $u$ be connected to $v$ through edge $(u, e_{i,uv}, v)$. We next describe how to produce observation embeddings $\boldsymbol{h}_{i,u}^t \in \mathbb{R}^{d_h}$ and $\boldsymbol{h}_{i,v}^t \in \mathbb{R}^{d_h}$ for sensors $u$ and $v$, respectively (Figure 3a). We omit layer index $l$ and note that the proposed strategy applies to any number of layers.

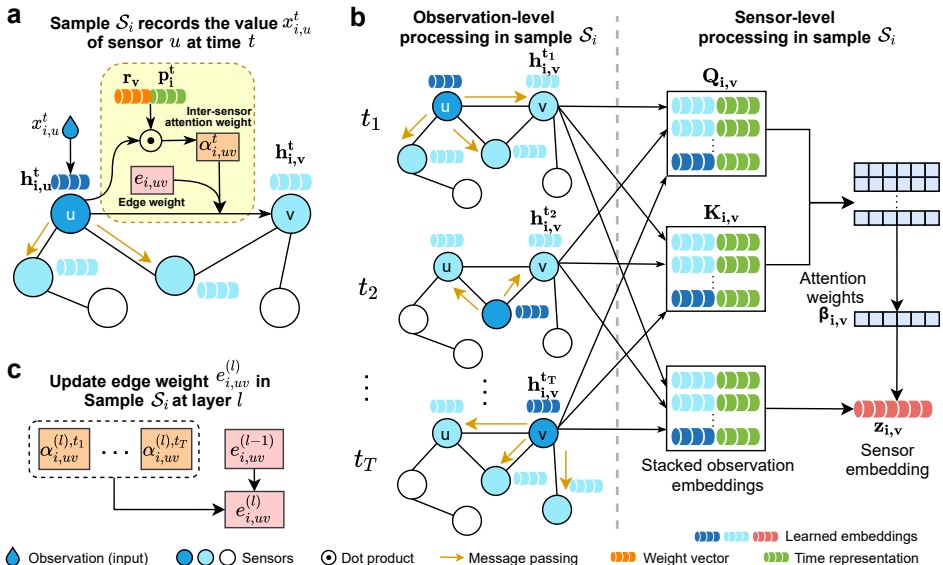

**Figure 3: (a)** RAINDROP generates observation embedding $\boldsymbol{h}_{i,u}^t$ based on observed value $x_{i,u}^t$ at $t$, passes message to neighbor sensors such as $v$, and generates $\boldsymbol{h}_{i,v}^t$ through inter-sensor dependencies. The $\alpha_{i,uv}^t$ denotes a time-specific attention weight, calculated based on time representation $\boldsymbol{p}_i^t$ and weight vector $\boldsymbol{r}_v$. Edge weight $e_{i,uv}$ is shared by all timestamps. **(b)** An illustration of generating sensor embedding. Apply the message passing in (a) to all timestamps and produce corresponding observation embeddings. We aggregate arbitrary number of observation embeddings into a fixed-length sensor embedding $\boldsymbol{z}_{i,v}$ while paying distinctive attentions to different observations. We independently apply the processing procedure to all sensors. **(c)** RAINDROP updates edge weight $e_{i,uv}^{(l)}$ based on the edge weight $e_{i,uv}^{(l-1)}$ from previous layer and the learned inter-sensor attention weights in all time steps. We explicitly show layer index $l$ as multiple layers are involved.

**Embedding an observation of an active sensor.** Let $u$ denote an active sensor whose value has just been observed as $x_{i,u}^t$. For sufficient expressive power (Veličković et al., 2018), we map observation $x_{i,u}^t$ to a high-dimensional space using a nonlinear transformation: $\boldsymbol{h}_{i,u}^t = \sigma(x_{i,u}^t \boldsymbol{R}_u)$. We use sensor-specific transformations because values recorded at different sensors can follow different distributions, which is achieved by trainable weight vectors $\boldsymbol{R}_u$ depending on what sensor is activated (Li et al., 2020b). Alternatives, such as a multilayer perceptron, can be considered to transform $x_{i,u}^t$ into $\boldsymbol{h}_{i,u}^t$. As $\boldsymbol{h}_{i,u}^t$ represents information brought on by observing $x_{i,u}^t$, we regard $\boldsymbol{h}_{i,u}^t$ as the embedding of $u$'s observation at $t$. Sensor-specific weight vectors $\boldsymbol{R}_u$ are shared across samples.

**Passing messages along sensor dependency graphs.** For sensors that are not active at timestamp $t$ but are neighbors of the active sensor $u$ in the sensor dependency graph $\mathcal{G}_i$, RAINDROP uses relationships between $u$ and those sensors to estimate observation embeddings for them. We proceed by describing how RAINDROP generates observation embedding $\boldsymbol{h}_{i,v}^t$ for sensor $v$ assuming $v$ is a neighbor of $u$ in $\mathcal{G}_i$. Given $\boldsymbol{h}_{i,u}^t$ and edge $(u, e_{i,uv}, v)$, we first calculate inter-sensor attention weight $\alpha_{i,uv}^t \in [0,1]$, representing how important $u$ is to $v$ via the following equation:

$$\alpha_{i,uv}^t = \sigma(\boldsymbol{h}_{i,u}^t \boldsymbol{D}[\boldsymbol{r}_v \| \boldsymbol{p}_i^t]^T), \tag{1}$$

where $\boldsymbol{r}_v \in \mathbb{R}^{d_r}$ is a trainable weight vector that is specific to the sensor receiving the message (i.e., $\boldsymbol{h}_{i,u}^t$). Vector $\boldsymbol{r}_v$ allows the model to learn distinct attention weights for different edges going out from the same sensor $u$. Further, $\boldsymbol{p}_i^t \in \mathbb{R}^{d_t}$ is the time representation obtained by converting a 1-dimensional timestamp $t$ into a multi-dimensional vector $\boldsymbol{p}_i^t$ by passing $t$ through a series of trigonometric functions (Horn et al., 2020). See Appendix A.1 for details. RAINDROP uses $\boldsymbol{p}_i^t$ to calculate attention weights that are sensitive to time. Finally, $\boldsymbol{D}$ is a trainable weight matrix mapping $\boldsymbol{h}_{i,u}^t$ from $d_h$ dimensions to $(d_r + d_t)$ dimensions. Taken this together, we can estimate the embedding $\boldsymbol{h}_{i,v}^t$ for $u$'s neighbor $v$ as follows:

$$\boldsymbol{h}_{i,v}^t = \sigma(\boldsymbol{h}_{i,u}^t \boldsymbol{w}_u \boldsymbol{w}_v^T \alpha_{i,uv}^t e_{i,uv}), \tag{2}$$

where $\boldsymbol{w}_u, \boldsymbol{w}_v \in \mathbb{R}^{d_h}$ are trainable weight vectors shared across all samples. The $\boldsymbol{w}_u$ is specific to active sensor $u$ and $\boldsymbol{w}_v$ is specific to neighboring sensor $v$. In the above equation, $e_{i,uv}$ denotes edge

weight shared across all timestamps. The above message passing describes the processing of a single observation at a single timestamp. In case multiple sensors are active at time $t$ and connected with $v$, we normalize $\alpha_{i,uv}^t$ (with softmax function) across active sensors and aggregate messages at $v$.

Overall, RAINDROP produces observation embedding $\boldsymbol{h}_{i,v}^t$ for sensor $v$ through its relational connection with $u$, even though there is no direct measurement of $v$ at time $t$. These message passing operations are performed to adaptively and dynamically estimate missing observations in the embedding space based on recorded information and learned graph structure.

**Updating sensor dependency graphs.** We describe the update of edge weights and prune of graph structures in the situation that stacks multiple RAINDROP layers (Figure 3). Here we explicitly show layer index $l$ because multiple layers are involved in the computation. As no prior knowledge is assumed, we initialize the graph as all sensors connected with each other. However, the fully connected edges may bridge sensors that should be independent, which will introduce spurious correlations and prevent the model from paying attention to the truly important connections. Addressing this issue, RAINDROP automatically updates edge weights and prunes out less important edges. Based on the aggregated temporal influence driven by the inter-sensor attention weights $\alpha_{i,uv}^{(l),t}$, we update edge weights $e_{i,uv}^{(l)}$ in each layer $l \in \{1, \ldots, L\}$ by:

$$e_{i,uv}^{(l)} = \frac{e_{i,uv}^{(l-1)}}{|\mathcal{T}_{i,u}|} \sum_{t \in \mathcal{T}_{i,u}} \alpha_{i,uv}^{(l),t}, \tag{3}$$

where $\mathcal{T}_{i,u}$ denotes the set of all timestamps where there is message passes from $u$ to $v$. In particular, we set $e_{i,uv}^{(0)} = 1$ in the initialization of graph structures. We use $L = 2$ in all our experiments. In every layer, we order the estimated values $e_{i,uv}^{(l)}$ for all edges in sample $\mathcal{S}_i$ and prune bottom $K\%$ edges with smallest edge weights (Yang et al., 2021). Pruned edges are not re-added in later layers.

## 3.4 GENERATING SENSOR EMBEDDINGS

Next we describe how to aggregate observation embeddings into sensor embeddings $\boldsymbol{z}_{i,v}$, taking sensor $v$ as an example (Figure 3b). Previous step (Sec. 3.3) generates observation embeddings for every timestamp when either $v$ or $v$'s neighbor is observed. The observation embeddings at different timestamps have unequal importance to the the sensor embedding (Zerveas et al., 2021). We use the temporal attention weight (scalar) $\beta_{i,v}^t$ to represent the importance of observation embedding at $t$. We use $\mathcal{T}_{i,v} = \{t_1, t_2, \ldots, t_T\}$ to denote all the timestamps when a readout is observed in $v$ (we can directly generate $\boldsymbol{h}_{i,v}^t$) or in $v$'s neighbor (we can generate $\boldsymbol{h}_{i,v}^t$ through message passing). The $\beta_{i,v}^t$ is the corresponding element of vector $\boldsymbol{\beta}_{i,v}$ which include the temporal attention weights at all timestamps $t \in \mathcal{T}_{i,v}$.

We use temporal self-attention to calculate $\boldsymbol{\beta}_{i,v}$, which is different from the standard self-attention (Hu et al., 2020; Yun et al., 2019). The standard dot-product self-attention generates an *attention matrix* with dimension of $T \times T$ (where $T = |\mathcal{T}_{i,v}|$ can vary across samples) that has an attention weight for each pair of observation embeddings. In our case, we only need a single *attention vector* where each element denotes the temporal attention weight of an observation embedding when generating the sensor embedding. Thus, we modify the typical self-attention model to fit our case: using a trainable $\boldsymbol{s} \in \mathbb{R}^{T \times 1}$ to map the self-attention matrix ($\mathbb{R}^{T \times T}$) to $T$-dimensional vector $\boldsymbol{\beta}_{i,v}$ ($\mathbb{R}^{T \times 1}$) through matrix product (Appendix A.2).

The following steps describe how to generate sensor embeddings. We first concatenate observation embedding $\boldsymbol{h}_{i,v}^t$ with time representation $\boldsymbol{p}_i^t$ to include information of timestamp. Then, we stack the concatenated embeddings $[\boldsymbol{h}_{i,v}^t || \boldsymbol{p}_i^t]$ for all $t \in \mathcal{T}_{i,v}$ into a matrix $\boldsymbol{H}_{i,v}$. The $\boldsymbol{H}_{i,v}$ contains all information of observations and timestamps for sensor $v$. We calculate $\beta_{i,v}^t$ through:

$$\boldsymbol{\beta}_{i,v} = \text{softmax}\left(\frac{\boldsymbol{Q}_{i,v}\boldsymbol{K}_{i,v}^T}{\sqrt{d_k}}\boldsymbol{s}\right), \tag{4}$$

where $\boldsymbol{Q}_{i,v}$ and $\boldsymbol{K}_{i,v}$ are two intermediate matrices that are derived from the stacked observation embeddings. In practice, $\boldsymbol{Q}_{i,v} = \boldsymbol{H}_{i,v}\boldsymbol{W}_Q$ and $\boldsymbol{K}_{i,v} = \boldsymbol{H}_{i,v}\boldsymbol{W}_K$ are linearly mapped from $\boldsymbol{H}_{i,v}$

parameterized by $\boldsymbol{W}_Q$ and $\boldsymbol{W}_K$, respectively (Vaswani et al., 2017). The $\sqrt{d_k}$ is a scaling factor where $d_k$ is the dimension after linear mapping. Based on the learned temporal attention weights $\beta_{i,v}^t$, we calculate sensor embedding $\boldsymbol{z}_{i,v}$ through:

$$\boldsymbol{z}_{i,v} = \sum_{t \in \mathcal{T}_{i,v}} (\beta_{i,v}^t [\boldsymbol{h}_{i,v}^t || \boldsymbol{p}_i^t] \boldsymbol{W}), \qquad (5)$$

where weight matrix $\boldsymbol{W}$ is a linear projector shared by all sensors and samples. It is worth to mention that all attention weights (such as $\alpha_{i,uv}^t$ and $\boldsymbol{\beta}_{i,v}$) can be multi-head. In this work, we describe the model in the context of single head for brevity.

Using attentional aggregation, RAINDROP can learn a fixed-length sensor embedding for arbitrary number of observations. Meanwhile, RAINDROP is capable of focusing on the most informative observation embeddings. We process all observation embeddings as a whole instead of sequentially, which allows parallel computation for faster training and also mitigates the performance drop caused by modeling long dependencies sequentially. In the case of sensors with very large number of observations, we can reduce the length of time series by subsampling or splitting a long series into multiple short series.

### 3.5 GENERATING SAMPLE EMBEDDINGS

Finally, for sample $\mathcal{S}_i$, we aggregate sensor embeddings $\boldsymbol{z}_{i,v}$ (Eq. 5) across all sensors to obtain an embedding $\boldsymbol{z}_i \in \mathbb{R}^{d_z}$ through a readout function $g$ as follows: $\boldsymbol{z}_i = g(\boldsymbol{z}_{i,v} \mid v = 1, 2, \ldots, M)$ (such as concatenation). When a sample contains a large number of sensors, RAINDROP can seamlessly use a set-based readout function such as averaging aggregation (Appendix A.3). Given an input sample $\mathcal{S}_i$, RAINDROP's strategy outlined in Sec. 3.2-3.5 produces a sample embedding $\boldsymbol{z}_i$ that can be further optimized for downstream tasks.

### 3.6 IMPLEMENTATION AND PRACTICAL CONSIDERATIONS

**Loss function.** RAINDROP's loss function is formulated as: $\mathcal{L} = \mathcal{L}_{\text{CE}} + \lambda \mathcal{L}_r$, where $\mathcal{L}_r = \frac{1}{M^2} \sum_{u,v \in \mathcal{V}} \sum_{i,j \in \mathcal{V}} ||e_{i,uv} - e_{j,uv}||_2 / (N-1)^2$, where $\mathcal{L}_{\text{CE}}$ is cross entropy and $\mathcal{L}_r$ is a regularizer to encourage the model to learn similar sensor dependency graphs for similar samples. The $\mathcal{L}_r$ measures averaged Euclidean distance between edge weights across all samples pairs, in all sensor pairs (including self-connections). The $\lambda$ is a user-defined coefficient. Practically, as $N$ can be large, we calculate $\mathcal{L}_r$ only for samples in a batch.

**Downstream tasks.** If a sample has auxiliary attributes (*e.g.*, a patient's demographics) that do not change over time, we can project the attribute vector to a $d_a$-dimensional vector $\boldsymbol{a}_i$ with a fully-connected layer and concatenate it with the sample embedding, getting $[\boldsymbol{z}_i || \boldsymbol{a}_i]$. At last, we feed $[\boldsymbol{z}_i || \boldsymbol{a}_i]$ (or only $\boldsymbol{z_i}$ if $\boldsymbol{a}_i$ is not available) into a neural classifier $\varphi : \mathbb{R}^{d_z + d_a} \to \{1, \ldots, C\}$. In our experiments, $\varphi$ is a 2-layer fully-connected network with $C$ neurons at the output layer returning prediction $\hat{y}_i = \varphi([\boldsymbol{z}_i || \boldsymbol{a}_i])$ for sample $\mathcal{S}_i$.

**Sensor dependencies.** While modeling sensor dependencies, we involve observation embedding ($\boldsymbol{h}_{i,u}^t$, Eq. 1) of each sample in the calculation of attention weights. Similarly, to model time-wise specificity in graph structures, we consider time information ($\boldsymbol{p}_i^t$, Eq. 1) when measuring $\alpha_{i,uv}^t$. RAINDROP can capture similar graph structures across samples from three aspects (Appendix A.4): (1) the initial graphs are the same in all samples; (2) the parameters in message passing ($\boldsymbol{R}_u$; $\boldsymbol{w}_u$, $\boldsymbol{w}_v$, Eq. 2), inter-sensor attention weights calculation ($\boldsymbol{D}$, Eq. 1), and temporal attention weights calculation ($s$, Eq. 4; $\boldsymbol{W}$, Eq. 5) are shared by all samples; (3) we encourage the model to learn similar graph structures by adding a penalty to disparity of structures ($\mathcal{L}_r$).

**Scalability.** RAINDROP is efficient because embeddings can be learned in parallel. In particular, processing of observation embeddings is independent across timestamps. Similarly, sensor embeddings can be processed independently across different sensors (Figure 3). While the complexity of temporal self-attention calculation grows quadratically with the number of observations, it can be practically implemented using highly-optimized matrix multiplication.

## 4 EXPERIMENTS

**Datasets.** Below we briefly overview healthcare and human activity datasets. (1) **P19** (Reyna et al., 2020) includes 38,803 patients that are monitored by 34 sensors. Each patient is associated with a binary label representing the occurrence of sepsis. (2) **P12** (Goldberger et al., 2000) records temporal measurements of 36 sensors of 11,988 patients in the first 48-hour stay in ICU. The samples are labeled based on hospitalization length. (3) **PAM** (Reiss & Stricker, 2012) contains 5,333 segments from 8 activities of daily living that are measured by 17 sensors. Details are in Appendix A.5.

**Baselines.** We compare RAINDROP with five state-of-the-art baselines: *Transformer* (Vaswani et al., 2017), *Trans-mean*, *GRU-D* (Che et al., 2018), *SeFT* (Horn et al., 2020), and *mTAND* (Shukla & Marlin, 2021). The *Trans-mean* is an imputation method combining transformer architecture with commonly used average interpolation (*i.e.*, missing values are replaced by average observations in each sensor). The *mTAND* (Shukla & Marlin, 2021) method has been shown to outperform numerous recurrent models including *RNN-Impute* (Che et al., 2018), *RNN-Simple*, and *Phased-LSTM* (Neil et al., 2016), along with ordinary differential equations (ODE)-based models such as *LATENT-ODE* and *ODE-RNN* (Chen et al., 2018). For this reason, we compare with *mTAND* and do not report comparison with those techniques in this paper. Even though, to better show the superiority of RAINDROP, we provide extensive comparison with popular approaches, such as DGM$^2$-O (Wu et al., 2021) and MTGNN (Wu et al., 2020c), that are designed for forecasting tasks. Further details are in Table 1 and Appendix A.11. Details on hyperparameter selection and baselines are in Appendix A.6, and evaluation metrics are presented in Appendix A.7.

### 4.1 RESULTS ACROSS DIVERSE EVALUATION SETTINGS

**Setting 1: Classic time series classification.** Setup. We randomly split the dataset into training (80%), validation (10%), and test (10%) set. The indices of these splits are fixed across all methods. Results. As shown in Table 1, RAINDROP obtains the best performance across three benchmark datasets, suggesting its strong performance for time series classification. In particular, in binary classification (P19 and P12), RAINDROP outperforms the strongest baselines by 5.3% in AUROC and 4.8% in AUPRC on average. In a more challenging 8-way classification on the PAM dataset, RAINDROP outperforms existing approaches by 5.7% in accuracy and 5.5% in F1 score. Further exploratory analyses and benchmarking results are shown in Appendix A.9-A.10.

**Table 1:** Method benchmarking on irregularly sampled time series classification (Setting 1).

| Methods | P19 | | P12 | | PAM | | | |
|---|---|---|---|---|---|---|---|---|
| | AUROC | AUPRC | AUROC | AUPRC | Accuracy | Precision | Recall | F1 score |
| Transformer | $83.2 \pm 1.3$ | $47.6 \pm 3.8$ | $65.1 \pm 5.6$ | $95.7 \pm 1.6$ | $83.5 \pm 1.5$ | $84.8 \pm 1.5$ | $86.0 \pm 1.2$ | $85.0 \pm 1.3$ |
| Trans-mean | $84.1 \pm 1.7$ | $47.4 \pm 1.4$ | $66.8 \pm 4.2$ | $95.9 \pm 1.1$ | $83.7 \pm 2.3$ | $84.9 \pm 2.6$ | $86.4 \pm 2.1$ | $85.1 \pm 2.4$ |
| GRU-D | $83.9 \pm 1.7$ | $46.9 \pm 2.1$ | $67.2 \pm 3.6$ | $95.9 \pm 2.1$ | $83.3 \pm 1.6$ | $84.6 \pm 1.2$ | $85.2 \pm 1.6$ | $84.8 \pm 1.2$ |
| SeFT | $78.7 \pm 2.4$ | $31.1 \pm 2.8$ | $66.8 \pm 0.8$ | $96.2 \pm 0.2$ | $67.1 \pm 2.2$ | $70.0 \pm 2.4$ | $68.2 \pm 1.5$ | $68.5 \pm 1.8$ |
| mTAND | $80.4 \pm 1.3$ | $32.4 \pm 1.8$ | $65.3 \pm 1.7$ | $96.5 \pm 1.2$ | $74.6 \pm 4.3$ | $74.3 \pm 4.0$ | $79.5 \pm 2.8$ | $76.8 \pm 3.4$ |
| IP-Net | $84.6 \pm 1.3$ | $38.1 \pm 3.7$ | $72.5 \pm 2.4$ | $96.7 \pm 0.3$ | $74.3 \pm 3.8$ | $75.6 \pm 2.1$ | $77.9 \pm 2.2$ | $76.6 \pm 2.8$ |
| DGM$^2$-O | $86.7 \pm 3.4$ | $44.7 \pm 11.7$ | $71.2 \pm 2.5$ | $96.9 \pm 0.4$ | $82.4 \pm 2.3$ | $85.2 \pm 1.2$ | $83.9 \pm 2.3$ | $84.3 \pm 1.8$ |
| MTGNN | $81.9 \pm 6.2$ | $39.9 \pm 8.9$ | $67.5 \pm 3.1$ | $96.4 \pm 0.7$ | $83.4 \pm 1.9$ | $85.2 \pm 1.7$ | $86.1 \pm 1.9$ | $85.9 \pm 2.4$ |
| **RAINDROP** | $\mathbf{87.0 \pm 2.3}$ | $\mathbf{51.8 \pm 5.5}$ | $\mathbf{72.1 \pm 1.3}$ | $\mathbf{97.0 \pm 0.4}$ | $\mathbf{88.5 \pm 1.5}$ | $\mathbf{89.9 \pm 1.5}$ | $\mathbf{89.9 \pm 0.6}$ | $\mathbf{89.8 \pm 1.0}$ |

**Setting 2: Leave-fixed-sensors-out.** Setup. RAINDROP can compensate for missing sensor observations by exploiting dependencies between sensors. To this end, we test whether RAINDROP can achieve good performance when a subset of sensors are completely missing. This setting is practically relevant in situations when, for example, sensors fail or are unavailable. We select a fraction of sensors and hide all their observations in both validation and test sets (training samples are not changed). In particular, we leave out the most informative sensors as defined by information gain analysis (Appendix A.8). The left-out sensors are fixed across samples and models. Results. We report results taking PAM as an example. In Table 2 (left block), we observe that RAINDROP achieves top performance in 18 out of 20 settings when the number of left-out sensors goes from 10% to 50%. With the increased amount of missing data, RAINDROP yield greater performance improvements. RAINDROP outperforms baselines by up to 24.9% in accuracy, 50.3% in precision, 29.3% in recall, and 42.8% in F1 score.

**Setting 3: Leave-random-sensors-out.** Setup. Setting 3 is similar to Setting 2 except that left-out sensors are randomly selected in each sample instead of being fixed. In each test sample, we select a subset of sensors and regard them as missing by replacing all of their observations with zeros. Results. We provide results for the PAM dataset in Table 2 (right block). We find that RAINDROP achieves better performance than baselines in 16 out of 20 settings and that Trans-mean and GRU-D are the strongest competitors. Further, we evaluated RAINDROP in another setting where the model is trained on one group of samples (*e.g.*, females) and tested on another group not seen during training (*e.g.*, males). Experimental setup and results are detailed in Appendix A.13.

**Table 2:** Classification performance on samples with a fixed set of left-out sensors (Setting 2) or random missing sensors (Setting 3) on the PAM dataset. Results for P19 dataset (Settings 2-3) are shown in Appendix A.12.

| Missing sensor ratio | Methods | PAM (Setting 2: leave-**fixed**-sensors-out) | | | | PAM (Setting 3: leave-**random**-sensors-out) | | | |
|---|---|---|---|---|---|---|---|---|---|
| | | Accuracy | Precision | Recall | F1 score | Accuracy | Precision | Recall | F1 score |
| 10% | Transformer | $60.3 \pm 2.4$ | $57.8 \pm 9.3$ | $59.8 \pm 5.4$ | $57.2 \pm 8.0$ | $60.9 \pm 12.8$ | $58.4 \pm 18.4$ | $59.1 \pm 16.2$ | $56.9 \pm 18.9$ |
| | Trans-mean | $60.4 \pm 11.2$ | $61.8 \pm 14.9$ | $60.2 \pm 13.8$ | $58.0 \pm 15.2$ | $62.4 \pm 3.5$ | $59.6 \pm 7.2$ | $63.7 \pm 8.1$ | $62.7 \pm 6.4$ |
| | GRU-D | $65.4 \pm 1.7$ | $72.6 \pm 2.6$ | $64.3 \pm 5.3$ | $63.6 \pm 0.4$ | $68.4 \pm 3.7$ | $74.2 \pm 3.0$ | $70.8 \pm 4.2$ | $72.0 \pm 3.7$ |
| | SeFT | $58.9 \pm 2.3$ | $62.5 \pm 1.8$ | $59.6 \pm 2.6$ | $59.6 \pm 2.6$ | $40.0 \pm 1.9$ | $40.8 \pm 3.2$ | $41.0 \pm 0.7$ | $39.9 \pm 1.5$ |
| | mTAND | $58.8 \pm 2.7$ | $59.5 \pm 5.3$ | $64.4 \pm 2.9$ | $61.8 \pm 4.1$ | $53.4 \pm 2.0$ | $54.8 \pm 2.7$ | $57.0 \pm 1.9$ | $55.9 \pm 2.2$ |
| | RAINDROP | $\mathbf{77.2 \pm 2.1}$ | $\mathbf{82.3 \pm 1.1}$ | $\mathbf{78.4 \pm 1.9}$ | $\mathbf{75.2 \pm 3.1}$ | $\mathbf{76.7 \pm 1.8}$ | $\mathbf{79.9 \pm 1.7}$ | $\mathbf{77.9 \pm 2.3}$ | $\mathbf{78.6 \pm 1.8}$ |
| 20% | Transformer | $63.1 \pm 7.6$ | $71.1 \pm 7.1$ | $62.2 \pm 8.2$ | $63.2 \pm 8.7$ | $62.3 \pm 11.5$ | $65.9 \pm 12.7$ | $61.4 \pm 13.9$ | $61.8 \pm 15.6$ |
| | Trans-mean | $61.2 \pm 3.0$ | $\mathbf{74.2 \pm 1.8}$ | $63.5 \pm 4.4$ | $64.1 \pm 4.1$ | $56.8 \pm 4.1$ | $59.4 \pm 3.4$ | $53.2 \pm 3.9$ | $55.3 \pm 3.5$ |
| | GRU-D | $64.6 \pm 1.8$ | $73.3 \pm 3.6$ | $63.5 \pm 4.6$ | $64.8 \pm 3.6$ | $64.8 \pm 0.4$ | $69.8 \pm 0.8$ | $65.8 \pm 0.5$ | $67.2 \pm 0.0$ |
| | SeFT | $35.7 \pm 0.5$ | $42.1 \pm 4.8$ | $38.1 \pm 1.3$ | $35.0 \pm 2.2$ | $34.2 \pm 2.8$ | $34.9 \pm 5.2$ | $34.6 \pm 2.1$ | $33.3 \pm 2.7$ |
| | mTAND | $33.2 \pm 5.0$ | $36.9 \pm 3.7$ | $37.7 \pm 3.7$ | $37.3 \pm 3.4$ | $45.6 \pm 1.6$ | $49.2 \pm 2.1$ | $49.0 \pm 1.6$ | $49.0 \pm 1.0$ |
| | RAINDROP | $\mathbf{66.5 \pm 4.0}$ | $72.0 \pm 3.9$ | $\mathbf{67.9 \pm 5.8}$ | $\mathbf{65.1 \pm 7.0}$ | $\mathbf{71.3 \pm 2.5}$ | $\mathbf{75.8 \pm 2.2}$ | $\mathbf{72.5 \pm 2.0}$ | $\mathbf{73.4 \pm 2.1}$ |
| 30% | Transformer | $31.6 \pm 10.0$ | $26.4 \pm 9.7$ | $24.0 \pm 10.0$ | $19.0 \pm 12.8$ | $52.0 \pm 11.9$ | $55.2 \pm 15.3$ | $50.1 \pm 13.3$ | $48.4 \pm 18.2$ |
| | Trans-mean | $42.5 \pm 8.6$ | $45.3 \pm 9.6$ | $37.0 \pm 7.9$ | $33.9 \pm 8.2$ | $\mathbf{65.1 \pm 1.9}$ | $63.8 \pm 1.2$ | $\mathbf{67.9 \pm 1.8}$ | $\mathbf{64.9 \pm 1.7}$ |
| | GRU-D | $45.1 \pm 2.9$ | $51.7 \pm 6.2$ | $42.1 \pm 6.6$ | $47.2 \pm 3.9$ | $58.0 \pm 2.0$ | $63.2 \pm 1.7$ | $58.2 \pm 3.1$ | $59.3 \pm 3.5$ |
| | SeFT | $32.7 \pm 2.3$ | $27.9 \pm 2.4$ | $34.5 \pm 3.0$ | $28.0 \pm 1.4$ | $31.7 \pm 1.5$ | $31.0 \pm 2.7$ | $32.0 \pm 1.2$ | $28.0 \pm 1.6$ |
| | mTAND | $27.5 \pm 4.5$ | $31.2 \pm 7.3$ | $30.6 \pm 4.0$ | $30.8 \pm 5.6$ | $34.7 \pm 5.5$ | $43.4 \pm 4.0$ | $36.3 \pm 4.7$ | $39.5 \pm 4.4$ |
| | RAINDROP | $\mathbf{52.4 \pm 2.8}$ | $\mathbf{60.9 \pm 3.8}$ | $\mathbf{51.3 \pm 7.1}$ | $\mathbf{48.4 \pm 1.8}$ | $60.3 \pm 3.5$ | $\mathbf{68.1 \pm 3.1}$ | $60.3 \pm 3.6$ | $61.9 \pm 3.9$ |
| 40% | Transformer | $23.0 \pm 3.5$ | $7.4 \pm 6.0$ | $14.5 \pm 2.6$ | $6.9 \pm 2.6$ | $43.8 \pm 14.0$ | $44.6 \pm 23.0$ | $40.5 \pm 15.9$ | $40.2 \pm 20.1$ |
| | Trans-mean | $25.7 \pm 2.5$ | $9.1 \pm 2.3$ | $18.5 \pm 1.4$ | $9.9 \pm 1.1$ | $48.7 \pm 2.7$ | $55.8 \pm 2.6$ | $54.2 \pm 3.0$ | $55.1 \pm 2.9$ |
| | GRU-D | $46.4 \pm 2.5$ | $\mathbf{64.5 \pm 6.8}$ | $42.6 \pm 7.4$ | $44.3 \pm 7.9$ | $47.7 \pm 1.4$ | $63.4 \pm 1.6$ | $44.5 \pm 0.5$ | $47.5 \pm 0.0$ |
| | SeFT | $26.3 \pm 0.9$ | $29.9 \pm 4.5$ | $27.3 \pm 1.6$ | $22.3 \pm 1.9$ | $26.8 \pm 2.6$ | $24.1 \pm 3.4$ | $28.0 \pm 1.2$ | $23.3 \pm 3.0$ |
| | mTAND | $19.4 \pm 4.5$ | $15.1 \pm 4.4$ | $20.2 \pm 3.8$ | $17.0 \pm 3.4$ | $23.7 \pm 1.0$ | $33.9 \pm 6.5$ | $26.4 \pm 1.6$ | $29.3 \pm 1.9$ |
| | RAINDROP | $\mathbf{52.5 \pm 3.7}$ | $53.4 \pm 5.6$ | $\mathbf{48.6 \pm 1.9}$ | $\mathbf{44.7 \pm 3.4}$ | $\mathbf{57.0 \pm 3.1}$ | $\mathbf{65.4 \pm 2.7}$ | $\mathbf{56.7 \pm 3.1}$ | $\mathbf{58.9 \pm 2.5}$ |
| 50% | Transformer | $21.4 \pm 1.8$ | $2.7 \pm 0.2$ | $12.5 \pm 0.4$ | $4.4 \pm 0.3$ | $43.2 \pm 2.5$ | $52.0 \pm 2.5$ | $36.9 \pm 3.1$ | $41.9 \pm 3.2$ |
| | Trans-mean | $21.3 \pm 1.6$ | $2.8 \pm 0.4$ | $12.5 \pm 0.7$ | $4.6 \pm 0.2$ | $46.4 \pm 1.4$ | $59.1 \pm 3.2$ | $43.1 \pm 2.2$ | $46.5 \pm 3.1$ |
| | GRU-D | $37.3 \pm 2.7$ | $29.6 \pm 5.9$ | $32.8 \pm 4.6$ | $26.6 \pm 5.9$ | $\mathbf{49.7 \pm 1.2}$ | $52.4 \pm 0.3$ | $42.5 \pm 1.7$ | $47.5 \pm 1.2$ |
| | SeFT | $24.7 \pm 1.7$ | $15.9 \pm 2.7$ | $25.3 \pm 2.6$ | $18.2 \pm 2.4$ | $26.4 \pm 1.4$ | $23.0 \pm 2.9$ | $27.5 \pm 0.4$ | $23.5 \pm 1.8$ |
| | mTAND | $16.9 \pm 3.1$ | $12.6 \pm 5.5$ | $17.0 \pm 1.6$ | $13.9 \pm 4.0$ | $20.9 \pm 3.1$ | $35.1 \pm 6.1$ | $23.0 \pm 3.2$ | $27.7 \pm 3.9$ |
| | RAINDROP | $\mathbf{46.6 \pm 2.6}$ | $\mathbf{44.5 \pm 2.6}$ | $\mathbf{42.4 \pm 3.9}$ | $\mathbf{38.0 \pm 4.0}$ | $47.2 \pm 4.4$ | $\mathbf{59.4 \pm 3.9}$ | $44.8 \pm 5.3$ | $47.6 \pm 5.2$ |

## 4.2 ABLATION STUDY AND VISUALIZATION OF OPTIMIZED SENSOR GRAPHS

**Ablation study.** Considering the PAM dataset and a typical setup (Setting 1), we conduct an ablation study to evaluate how much various RAINDROP's components contribute towards its final performance. We examine the following components: inter-sensor dependencies (further decomposed into weights including $e_{i,uv}$, $\boldsymbol{r}_v$, $\boldsymbol{p}_i^t$, and $\alpha_{i,uv}^t$), temporal attention, and sensor-level concatenation. We show in Appendix A.14 (Table 7) that all model components are necessary and that regularization $\mathcal{L}_r$ contributes positively to RAINDROP's performance.

**Visualizing sensor dependency graphs.** We investigate whether samples with the same labels get more similar sensor dependency graphs than samples with different labels. To this end, we visualize inter-sensor dependencies (P19; Setting 1) and explore them. Figure 4 shows distinguishable patterns between graphs of negative and positive samples, indicating that RAINDROP can extract relationships that are specific to downstream sample labels. Further differential analysis provides insights that can inform early detection of sepsis from P19 clinical data. Details are in Appendix A.15.

## 5 CONCLUSION

We introduce RAINDROP, a graph-guided network for irregularly sampled time series. RAINDROP learns a distinct sensor dependency graph for every sample capturing time-varying dependencies between sensors. The ability to leverage graph structure gives RAINDROP unique capability to naturally handle misaligned observations, non-uniform time intervals between successive observations, and sensors with varying numbers of recorded observations. Our findings have implications for using message passing as a way to leverage relational information in multivariate time series.

## ACKNOWLEDGMENTS

This material is based upon work supported by the Under Secretary of Defense for Research and Engineering under Air Force Contract No. FA8702-15-D-0001. M.Z. is supported, in part, by NSF under nos. IIS-2030459 and IIS-2033384, Harvard Data Science Initiative, Amazon Research Award, Bayer Early Excellence in Science Award, AstraZeneca Research, and Roche Alliance with Distinguished Scientists Award. Any opinions, findings, conclusions or recommendations expressed in this material are those of the authors and do not necessarily reflect the views of the funders. The authors declare that there are no conflict of interests.

## REPRODUCIBILITY STATEMENT

We ensure the reproducibility of our work by clearly presenting the model and providing publicly accessible code and data. For all datasets used in this work, we share downloadable links to the raw sources and processed and ready-to-run datasets with the research community through this link: https://github.com/mims-harvard/Raindrop. We specify all training details (e.g., preprocessing, data splits, hyperparameters, sensor selection) in the main text and Appendix. Python implementation of RAINDROP and all baseline methods is available at the aforementioned link. Detailed description of data, scripts, and configurations along with examples of usage are also provided.

## ETHICS STATEMENT

The ability of RAINDROP to learn robust information about sensors' representations and dependencies creates new opportunities for applications, where time series are predominant, *e.g.*, in healthcare, biology, and finance. In all these fields, especially in healthcare applications, our method should be used with caution. Although our model can gain valuable insights from time series, users must consider the limitations of machine-guided predictions. As with all data-driven solutions, our model may make biased predictions. In the case of biomedical data, biases can exist within the data itself, which can be, for example, caused by considering demographic attributes, such as age, weight, and gender, that might correlate with protected/regulated attributes. When target classes are highly imbalanced, our model can mitigate the issues by upsampling minority classes in every processed batch.

All datasets in this paper are publicly available and are not associated with any privacy or security concern. Further, all data are anonymized to guard against breaching patients' protected health information. We followed PhysioNet privacy policy and guidelines (https://archive.physionet.org/privacy.shtml) when experimenting with P12 and P19 datasets.

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

# A APPENDIX

## A.1 ENCODING TIMESTAMPS

For a given time value $t$, we pass it to trigonometric functions with the frequency of 10,000 (Vaswani et al., 2017) and generate time representation $\boldsymbol{p}^t \in \mathbf{R}^\xi$ (omit sample index $i$ for brevity) through (Horn et al., 2020):

$$\boldsymbol{p}_{2k}^t = \sin(\frac{t}{10000^{2k/\xi}}), \quad \boldsymbol{p}_{2k+1}^t = \cos(\frac{t}{10000^{2k/\xi}}), \tag{6}$$

where $\xi$ is the expected dimension. In this work, we set $\xi = 16$ in all experimental settings for all models. Please note, we encode the *time value* which is a continuous timestamp, instead of *time position* which is a discrete integer indicating the order of observation in time series.

## A.2 ADDITIONAL INFORMATION ON THE CALCULATION OF TEMPORAL ATTENTION WEIGHT

The Eq. 4 describes how we learn the temporal attention weights vector $\boldsymbol{\beta}_{i,v}$ for sensor $v$, following the self-attention formalism. Different from the standard self-attention mechanism that generates an self-attention matrix, we generate a temporal attention weight vector. The reason is that we only need an attention weight vector (instead of a matrix) to aggregate the observation embeddings into a single sensor embedding through weighted sum.

In the standard self-attention matrix, each element denotes the dependency of an observation embedding on another observation embedding. Similarly, each row describes the dependencies of an observation embedding on all other observation embeddings (all the observations belong to the same sensor). Our intuition is to aggregate a row in the self-attention matrix into a scalar that denotes the importance of the observation embedding to the whole sensor embedding.

In practice, we apply the weighted aggregation, parameterized by $\boldsymbol{s}$, to every row in the self-attention matrix and concatenate the generated scalars into an attention vector. Next, we give a concrete example to specifically describe the meaning of $\boldsymbol{s}$. Each row, $j$, of the self-attention matrix captures relationships of observation embedding $\boldsymbol{h}_{i,v}^{t_j}$ to all observation embeddings $\{\boldsymbol{h}_{i,v}^{t_k} : k = 1, ..., T\}$. Then, using the learnable weight vector $\boldsymbol{s}$, these correlations between observations are aggregated across time to obtain temporal importance weight $\beta_{i,v}^{t_j}$. The $\beta_{i,v}^{t_j}$ represents the importance of the corresponding observation to the whole sensor embedding.

## A.3 ADDITIONAL INFORMATION ON SAMPLE EMBEDDING

As we generate sample embedding by concatenating all sensor embeddings, the sample embedding could be relatively long when there is a large number of sensors. To alleviate this issue, on one hand, we can reduce the dimension of sample embeddings by adding a neural layer (such as a simple fully-connected layer) after the concatenation. On the other hand, when the number of sensors is super large, our model is flexible and can effortlessly switch the concatenation to other readout functions (such as averaging aggregation): this will naturally solve the problem of long vectors. We empirically show that concatenation works better than averaging in our case. We see a boost in the AUROC score by 0.6% using concatenation instead of averaging for generating sample embeddings(P19; Setting 1).

## A.4 ADDITIONAL INFORMATION ON SAMPLE SIMILARITIES

In this work, we assume all samples share some common characteristics to some extent. When modeling the similarities across samples, we do not consider the situation where the samples are similar within latent groups and different across groups.

Our study focuses on the question of irregularity rather than the question of distribution shifts in time series. To this end, in our experiments, we first rigorously benchmark Raindrop using a standard evaluating setup (Setting 1, which is classification of irregular time series). This is the only setup that most existing methods consider (e.g., Shukla & Marlin (2021); Che et al. (2018)) and we want to make sure our comparisons are fair. In order to provide a more rigorous assessment of Raindrop's performance, we also consider more challenging setups in our experiments (i.e., Settings 2-4) when the dataset is evaluated in a non-standard manner and the split is informed by a select data attribute.

**Table 3:** Dataset statistics. The '#-timestamps' refers to the number of all sampling timestamps measured in this dataset. The '#-classes' means the number of categories in dataset labels. The 'Static info' indicates if sample's static attributes (e.g., height and weight) are available. The 'missing ratio' denotes the ratio between the number of missing observations and the number of all possible observations if the dataset is fully-observed.

| Datasets | #-samples | #-sensors | #-timestamps | #-classes | Static info | Missing ratio (%) |
|---|---|---|---|---|---|---|
| P19 | 38,803 | 34 | 60 | 2 | True | 94.9 |
| P12 | 11,988 | 36 | 215 | 2 | True | 88.4 |
| PAM | 5,333 | 17 | 600 | 8 | False | 60.0 |

Our results on Setting 1 are consistent with those on Settings 2-4. Results on harder Settings 2-4 show that Raindrop can perform comparably better than baselines. Results across these diverse settings increase our confidence that Raindrop is quite flexible and widely applicable.

## A.5 FURTHER DETAILS ON DATASETS

**P19: PhysioNet Sepsis Early Prediction Challenge 2019.** P19 dataset (Reyna et al., 2020) contains 38,803 patients and each patient is monitored by 34 irregularly sampled sensors including 8 vital signs and 26 laboratory values. The original dataset has 40,336 patients, we remove the samples with too short or too long time series, remaining 38,803 patients (the longest time series of the patient has more than one and less than 60 observations). Each patient is associated with a static vector indicating attributes: age, gender, time between hospital admission and ICU admission, ICU type, and ICU length of stay (days). Each patient has a binary label representing occurrence of sepsis within the next 6 hours. The dataset is highly imbalanced with only ∼4% positive samples.

**P12: PhysioNet Mortality Prediction Challenge 2012.** P12 dataset (Goldberger et al., 2000) includes 11,988 patients (samples), after removing 12 inappropriate samples following (Horn et al., 2020). Each patient contains multivariate time series with 36 sensors (excluding weight), which are collected in the first 48-hour stay in ICU. Each sample has a static vector with 9 elements including age, gender, etc. Each patient is associated with a binary label indicating length of stay in ICU, where negative label means hospitalization is not longer than 3 days and positive label marks hospitalization is longer than 3 days. P12 is imbalanced with ∼93% positive samples.

**PAM: PAMAP2 Physical Activity Monitoring.** PAM dataset (Reiss & Stricker, 2012) measures daily living activities of 9 subjects with 3 inertial measurement units. We modify it to suit our scenario of irregular time series classification. We excluded the ninth subject due to short length of sensor readouts. We segment the continuous signals into samples with the time window of 600 and the overlapping rate of 50%. PAM originally has 18 activities of daily life. We exclude the ones associated with less than 500 samples, remaining 8 activities. After modification, PAM dataset contains 5,333 segments (samples) of sensory signals. Each sample is measured by 17 sensors and contains 600 continuous observations with the sampling frequency 100 Hz. To make time series irregular, we randomly remove 60% of observations. To keep fair comparison, the removed observations are randomly selected but kept the same for all experimental settings and approaches. PAM is labelled by 8 classes where each class represents an activity of daily living. PAM does not include static attributes and the samples are approximately balanced across all 8 categories.

To feed given data into neural networks, we set the input as zero if no value was measured. In highly imbalanced datasets (P19 and P12) we perform batch minority class upsampling, which means that every processed batch has the same number of positive and negative class samples. The dataset statistics including sparse ratio are provided in Table 3.

## A.6 FURTHER DETAILS ON MODEL HYPERPARAMETERS

**Baseline hyperparameters.** The implementation of baselines follows the corresponding papers including SeFT (Horn et al., 2020), GRU-D (Che et al., 2018), and mTAND (Shukla & Marlin, 2021). We follow the settings of Transformer baseline in (Horn et al., 2020) while implementing Transformer in our work. For average imputation in Trans-mean, we replace the missing values by the global mean value of observations in the sensor (Shukla & Marlin, 2020). We use batch size of

128 and learning rate of 0.0001. Note that we upsample the minority class in each batch to make the batch balance (64 positive samples and 64 negative samples in each batch).

The chosen hyperparameters are the same across datasets (P19, P12, PAM), models (both baselines and RAINDROP), and experimental settings. Remarkably, we found that all the baselines make dummy predictions (classify all testing samples as the majority label) on PAM in Setting 2-3 while RAINDROP makes reasonable predictions. For the comparison to make sense (*i.e.*, the baselines can make meaningful predictions), we use learning rate of 0.001 for baselines on PAM. GRU-D has 49 layers while other models have 2 layers. We run all models for 20 epochs, store the parameters that obtain the highest AUROC in the validation set, and use it to make predictions for testing samples. We use the Adam algorithm for gradient-based optimization (Kingma & Ba, 2014).

**RAINDROP hyperparameters.** Next, we report the setting of unique hyperparameters in our RAIN-DROP. In the generation of observation embedding, we set $\boldsymbol{R}_u$ as a 4-dimensional vector, thus the produced observation embedding has 4 dimensions. The dimensions of time representation $\boldsymbol{p}^t$ and $\boldsymbol{r}_v$ are both 16. The trainable weight matrix $\boldsymbol{D}$ has shape of $4 \times 32$. The dimensions of $\boldsymbol{w}_u$ and $\boldsymbol{w}_v$ are the same as the number of sensors: 34 in P19, 36 in P12, and 17 in PAM. We set the number of RAINDROP layers $L$ as 2 while the first layer prunes edges and the second layer does not. We set the proportion of edge pruning as 50% (K=50), which means we remove half of the existing edges that have the lowest weights. The $d_k$ is set to 20, while the shape of $\boldsymbol{W}$ is $20 \times 20$. All the activation functions, without specific clarification, are sigmoid functions. The $d_a$ is set equal to the number of sensors. The first layer of $\varphi$ has 128 neurons while the second layer has $C$ neurons (*i.e.*, 2 for P19 and P12; 8 for PAM). We set $\lambda = 0.02$ to adjust $\mathcal{L}_r$ regularization scale. All the preprocessed datasets and implementation codes are made available online. Further details are available through RAINDROP's code and dataset repository.

**Readout function.** Here we discuss the selection of readout function $g$ in section 3.5. Our preliminary experiments show that concatenation outperforms other popular aggregation functions such as averaging (Errica et al., 2021) and squeeze-excitation readout function (Kim et al., 2021; Hu et al., 2018). While any of those aggregation functions can be considered, we used concatenation throughout all experiments in this manuscript.

## A.7  PERFORMANCE METRICS

Since P19 and P12 datasets are imbalanced, we use the Area Under a ROC Curve (AUROC) and Area Under Precision-Recall Curve (AUPRC) to measure performance. As the PAM dataset is nearly balanced, we also report accuracy, precision, recall and F1 score. We report mean and standard deviation values over 5 independent runs. Model parameters that achieve the best AUROC value on the validation set are used for test set.

## A.8  FURTHER DETAILS ON SETUP DETAILS FOR SETTING 2

In Setting 2, the selected missing sensors are fixed across different models and chosen in the following way. First, we calculate the importance score for each sensor and rank them in a descending order. The importance score is based on information gain, which we calculate with feeding the observations into a Random Forest classifier with 20 decision trees. In particular, we treat each sample as only having one sensor, then feed the single sensor into random forest classifier and record the AUROC. The higher AUROC indicates the sensor provides higher information gain. When we have sensors ranked by their AUROC values, we choose the first $n$ sensors (the ones with highest AUROC values) and replace all observations in these sensors by zeros in all samples in validation and test set. The number of missing sensors is defined indirectly from the user with the sensors' missing ratio which ranges from 0.1 to 0.5.

## A.9  ADDITIONAL INFORMATION ON MISSING PATTERN

This work propose RAINDROP which is a novel solution for irregularity in multivariate time series through inter-sensor dependencies. RAINDROP is not in conflict with other solutions (such as missing pattern and temporal decay) for irregularity. However, as the missing pattern is widely discussed in modelling incomplete time series (Che et al., 2018), we explore how to combine the advantages of relational structures and missing pattern. We adopt mask matrix as a proxy of missing pattern

as in Che et al. (2018). Taking the architecture of RAINDROP, we concatenate the observation $x_{i,u}^t$ with a binary mask indicator $b_{i,u}^t$ as input. The indicator $b_{i,u}^t$ is set as 1 when there is an observation of sensor $i$ at time $t$ and set as 0 otherwise. All the experimental settings and hyperparameters are the same as in RAINDROP (P19; Setting 1). The experimental results show that taking advantage of missing pattern can slightly boost the AUROC by 1.2% and AUPRC by 0.9% in P19. This empirically shed the light for future research on integrating multiple characteristics in representation of irregularly time series.

## A.10 COMPARISON BETWEEN TEMPORAL ATTENTION AND LSTM

We conduct extensive experiments to compare the effectiveness of temporal attention and LSTM. To this end, we replace the temporal attention in sensor embedding generation (Eq 4-5) in RAINDROP by LSTM layer which processes all observation embeddings sequentially. We use zero padding to convert the irregular observations into fixed-length time series so the data can be fed into LSTM architecture. We regard the last output of LSTM as generated sensor embedding. The number of LSTM cells equal to the dimension of observation embedding. All the model structures are identical except in the part of temporal attention and LSTM. We keep all experimental settings (P19; Setting 1) and hyperparameter selections the same. The experimental results show that the temporal self-attention outperform LSTM by 1.8% (AUROC) and additionally saved 49% of the training time. One potential reason is that the self-attention mechanism avoids recursion and allows parallel computation and also reduces performance degradation caused by long-term dependencies (Ganesh et al., 2021; Vaswani et al., 2017).

## A.11 ADDITIONAL INFORMATION ON METHOD BENCHMARKING

Taking experimental Setting 1 (*i.e.*, classic time series classification) as an example, we conduct extensive experiments to compare Raindrop with ODE-RNN (Chen et al., 2020), DGM$^2$-O (Wu et al., 2021), EvoNet (Hu et al., 2021), and MTGNN (Wu et al., 2020c). As IP-Net (Shukla & Marlin, 2018) and mTAND (Shukla & Marlin, 2021) are from the same authors, we only compare with mTAND which is the latest model. For the baselines, we follow the settings as provided in their public codes. For methods, which cannot deal with irregular data (*e.g.*, EvoNet and MTGNN), we first impute the missing data using mean imputation and then feed data into the model. For forecasting models (*e.g.*, MTGNN) which are strictly not comparable with the proposed classification model, we formulate the task as a single-step forecasting, concatenate the learned representations from all sensors and feed into a fully-connected layer (work as classifier) to make prediction, and use cross-entropy to quantify the loss.

## A.12 RESULTS FOR P19 (SETTINGS 2-3)

Here we report the experimental results for P19 in Setting 2 (Table 4) and Setting 3 (Table 5).

**Table 4:** Classification on samples with fixed missing sensors (P19; Setting 2)

| Models | Missing ratio | | | | | | | | | | | |
|---|---|---|---|---|---|---|---|---|---|---|---|---|
| | 0% | | 10% | | 20% | | 30% | | 40% | | 50% | |
| | AUROC | AUPRC | AUROC | AUPRC | AUROC | AUPRC | AUROC | AUPRC | AUROC | AUPRC | AUROC | AUPRC |
| Transformer | 83.2 ± 1.3 | 47.6 ± 3.8 | 77.4 ± 3.5 | 38.2 ± 4.2 | 75.7 ± 3.4 | 35.2 ± 5.4 | 75.1 ± 3.5 | 35.5 ± 4.4 | 75.3 ± 3.5 | 36.2 ± 4.2 | 74.9 ± 3.1 | 35.5 ± 5.0 |
| Trans-mean | 84.1 ± 1.7 | 47.4 ± 1.4 | 79.2 ± 2.7 | 40.6 ± 5.7 | 79.8 ± 2.5 | 38.3 ± 2.8 | 76.9 ± 2.4 | 37.5 ± 5.9 | 76.4 ± 2.0 | 36.3 ± 5.8 | 74.1 ± 2.3 | 41.3 ± 4.7 |
| GRU-D | 83.9 ± 1.7 | 46.9 ± 2.1 | 79.6 ± 2.2 | 37.4 ± 2.5 | 77.5 ± 3.1 | 36.5 ± 4.6 | 76.6 ± 2.9 | 35.1 ± 2.4 | 74.6 ± 2.7 | 35.9 ± 2.7 | 74.1 ± 2.9 | 33.2 ± 3.8 |
| SeFT | 78.7 ± 2.4 | 31.1 ± 2.8 | 77.3 ± 2.4 | 25.5 ± 2.3 | 63.5 ± 2.0 | 14.0 ± 1.1 | 62.3 ± 2.1 | 12.9 ± 1.2 | 57.8 ± 1.7 | 9.8 ± 1.1 | 56.0 ± 3.1 | 7.8 ± 1.3 |
| mTAND | 80.4 ± 1.3 | 32.4 ± 1.8 | 79.7 ± 2.2 | 29.0 ± 4.3 | 77.8 ± 1.9 | 25.3 ± 2.4 | 77.7 ± 1.9 | 27.8 ± 2.6 | 79.4 ± 2.0 | 32.1 ± 2.1 | 77.3 ± 2.1 | 27.0 ± 2.5 |
| RAINDROP | **87.0 ± 2.3** | **51.8 ± 5.5** | **84.3 ± 2.5** | **46.1 ± 3.5** | **81.9 ± 2.1** | **45.2 ± 6.4** | **81.4 ± 2.1** | **43.7 ± 7.2** | **81.8 ± 2.2** | **44.9 ± 6.6** | **79.7 ± 1.9** | **43.8 ± 5.6** |

**Table 5:** Classification on samples with random missing sensors (P19; Setting 3)

| Models | Missing ratio | | | | | | | | | | | |
|---|---|---|---|---|---|---|---|---|---|---|---|---|
| | 0% | | 10% | | 20% | | 30% | | 40% | | 50% | |
| | AUROC | AUPRC | AUROC | AUPRC | AUROC | AUPRC | AUROC | AUPRC | AUROC | AUPRC | AUROC | AUPRC |
| Transformer | 83.2 ± 1.3 | 47.6 ± 3.8 | 82.2 ± 2.7 | 46.8 ± 3.5 | 81.6 ± 3.5 | 42.5 ± 8.5 | 81.3 ± 3.1 | 42.1 ± 4.5 | 80.2 ± 2.9 | 41.9 ± 6.8 | 79.2 ± 1.9 | 43.7 ± 3.7 |
| Trans-mean | 84.1 ± 1.7 | 47.4 ± 1.4 | 82.5 ± 3.7 | 44.7 ± 6.8 | 81.7 ± 2.0 | 45.9 ± 3.6 | 81.2 ± 2.2 | 43.2 ± 6.3 | 80.2 ± 1.7 | 41.5 ± 4.8 | 79.8 ± 3.1 | 39.3 ± 5.1 |
| GRU-D | 83.9 ± 1.7 | 46.9 ± 2.1 | 81.2 ± 3.4 | 46.4 ± 2.7 | 78.6 ± 4.1 | 43.3 ± 2.4 | 76.3 ± 2.5 | 28.5 ± 2.1 | 74.2 ± 2.7 | 29.6 ± 3.1 | 74.6 ± 3.5 | 26.5 ± 4.2 |
| SeFT | 78.7 ± 2.4 | 31.1 ± 2.8 | 76.8 ± 2.2 | 28.3 ± 2.5 | 77.0 ± 2.2 | 24.1 ± 2.4 | 75.2 ± 2.2 | 22.5 ± 3.0 | 73.6 ± 2.7 | 18.3 ± 3.2 | 72.6 ± 2.5 | 15.7 ± 1.9 |
| mTAND | 80.4 ± 1.3 | 32.4 ± 1.8 | 75.2 ± 2.5 | 24.5 ± 2.4 | 74.4 ± 3.5 | 24.6 ± 3.5 | 74.2 ± 3.2 | 22.6 ± 2.3 | 74.1 ± 2.6 | 23.1 ± 3.6 | 73.9 ± 3.7 | 24.6 ± 3.7 |
| RAINDROP | **87.0 ± 2.3** | **51.8 ± 5.5** | **85.5 ± 2.1** | **50.2 ± 5.5** | **83.5 ± 3.2** | **47.4 ± 7.0** | **83.1 ± 1.5** | **48.2 ± 4.7** | **82.6 ± 1.7** | **48.0 ± 5.5** | **80.9 ± 2.4** | **45.2 ± 6.9** |

**Table 6:** Comparison of results when excluding dependency graph in RAINDROP (P19; Setting 4). The results are the same as in Table 8 except the row of 'RAINDROP w/o graph', where we do not consider inter-sensor dependencies and set all sensors as independent in the dependency graph.

| Model | Generalizing to a new patient group | | | | | | | |
| --- | --- | --- | --- | --- | --- | --- | --- | --- |
| | Train: Young → Test: Old | | Train: Old → Test: Young | | Train: Male → Test: Female | | Train: Female → Test: Male | |
| | AUROC | AUPRC | AUROC | AUPRC | AUROC | AUPRC | AUROC | AUPRC |
| Transformer | $76.2 \pm 0.7$ | $30.5 \pm 4.8$ | $76.5 \pm 1.1$ | $33.7 \pm 5.7$ | $77.8 \pm 1.1$ | $26.0 \pm 6.2$ | $75.2 \pm 1.0$ | $30.3 \pm 5.5$ |
| Trans-mean | $80.6 \pm 1.4$ | $39.8 \pm 4.2$ | $78.4 \pm 1.1$ | $35.8 \pm 2.9$ | $80.2 \pm 1.7$ | $32.1 \pm 1.9$ | $76.4 \pm 0.8$ | $32.5 \pm 3.3$ |
| GRU-D | $76.5 \pm 1.7$ | $29.5 \pm 2.3$ | $79.6 \pm 1.7$ | $35.2 \pm 4.6$ | $78.5 \pm 1.6$ | $31.9 \pm 4.8$ | $76.3 \pm 2.5$ | $31.1 \pm 2.6$ |
| SeFT | $77.5 \pm 0.7$ | $26.6 \pm 1.2$ | $78.9 \pm 1.0$ | $32.7 \pm 2.7$ | $78.6 \pm 0.6$ | $31.1 \pm 1.2$ | $76.9 \pm 0.5$ | $26.4 \pm 1.1$ |
| mTAND | $79.0 \pm 0.8$ | $28.8 \pm 2.3$ | $79.4 \pm 0.6$ | $29.8 \pm 1.2$ | $78.0 \pm 0.9$ | $26.5 \pm 1.7$ | $78.9 \pm 1.2$ | $29.2 \pm 2.0$ |
| RAINDROP w/o graph | $80.5 \pm 1.1$ | $31.6 \pm 2.1$ | $78.5 \pm 0.9$ | $36.7 \pm 2.7$ | $81.3 \pm 1.5$ | $36.8 \pm 1.7$ | $77.5 \pm 1.9$ | $33.4 \pm 2.6$ |
| RAINDROP | $\mathbf{83.2 \pm 1.6}$ | $\mathbf{43.6 \pm 4.7}$ | $\mathbf{82.0 \pm 4.4}$ | $\mathbf{44.3 \pm 3.6}$ | $\mathbf{85.0 \pm 1.4}$ | $\mathbf{45.2 \pm 2.9}$ | $\mathbf{81.2 \pm 3.8}$ | $\mathbf{40.7 \pm 2.9}$ |

**Table 7:** Results of ablation study on the PAM dataset (Setting 1).

| RAINDROP Model | | Accuracy | Precision | Recall | F1 score |
| --- | --- | --- | --- | --- | --- |
| W/o weights vector $\boldsymbol{R}_u$ | | $81.1 \pm 2.6$ | $81.9 \pm 2.4$ | $80.1 \pm 1.6$ | $81.6 \pm 2.1$ |
| W/o inter-sensor dependency | W/o $e_{i,uv}$ | $82.6 \pm 1.2$ | $82.9 \pm 1.6$ | $84.3 \pm 1.4$ | $83.8 \pm 1.7$ |
| | W/o $\boldsymbol{r}_v$ | $86.5 \pm 2.4$ | $83.3 \pm 1.9$ | $82.6 \pm 1.5$ | $82.9 \pm 1.4$ |
| | W/o $\boldsymbol{p}_i^t$ | $79.8 \pm 2.7$ | $80.1 \pm 3.6$ | $80.6 \pm 1.7$ | $80.2 \pm 2.9$ |
| | W/o $\alpha_{i,uv}^t$ | $85.2 \pm 2.5$ | $86.4 \pm 2.7$ | $84.5 \pm 2.9$ | $85.6 \pm 2.9$ |
| W/o temporal attention | | $81.5 \pm 1.9$ | $84.6 \pm 1.7$ | $83.9 \pm 2.5$ | $84.2 \pm 2.2$ |
| W/o sensor level concatenation | | $84.4 \pm 2.1$ | $86.7 \pm 1.1$ | $85.2 \pm 1.9$ | $85.8 \pm 2.6$ |
| W/o regularization term $\mathcal{L}_r$ | | $87.3 \pm 2.9$ | $88.6 \pm 3.4$ | $87.1 \pm 2.8$ | $87.6 \pm 3.1$ |
| Full RAINDROP | | $\mathbf{88.5 \pm 1.5}$ | $\mathbf{89.9 \pm 1.5}$ | $\mathbf{89.9 \pm 0.6}$ | $\mathbf{89.8 \pm 1.0}$ |

## A.13 EVALUATION ON GROUP-WISE TIME SERIES CLASSIFICATION

To understand whether RAINDROP can adaptively adjust its structure and generalize well to other groups of samples which were not observed while training the model. In this setting we split the data into two groups, based on a specific static attribute. The first split attribute is *age*, where we classify people into *young* ($< 65$ years) and *old* ($\geq 65$ years) groups. We also split patients into *male* and *female* by *gender* attribute. Given the split attribute, we use one group as a train set and randomly split the other group into equally sized validation and test set.

Taking P19 as an example, we present the classification results when the training and testing samples are from different groups. As shown in Table 8, RAINDROP achieves the best results over all of the four given cross-group scenarios. For instance, RAINDROP claims large margins (with 4.8% in AUROC and 13.1% in AUPRC absolute improvement) over the second best model while training on males and testing on female patients.

Although RAINDROP is not designed to address domain adaptation explicitly, the results show that RAINDROP performs better than baselines when transferring from one group of samples to another. One reason for our good performance is that the learned inter-sensor weights and dependency graphs are sample-specific and their learning is based on the sample's observations. Thus, the proposed RAINDROP has the power, to some extent, to adaptively learn the inter-sensor dependencies based on the test sample's measurements. RAINDROP is not generalizing to new groups, but generalizing to new samples, which leads to a good performance even though our model is not designed for domain adaptation. We validate the reason empirically. We remove the inter-sensor dependencies (set all sensors isolated in the dependency graph; set all $\alpha_{i,uv}^t$ and $e_{i,uv}^t$ as 0) in RAINDROP and evaluate the model in group-wise time series classification. The experimental results show that the performance drops a lot when excluding dependency graphs and message passing in RAINDROP (Table 6). Without inter-sensor dependencies our model is on par with other baselines and does not outperform them by a large margin.

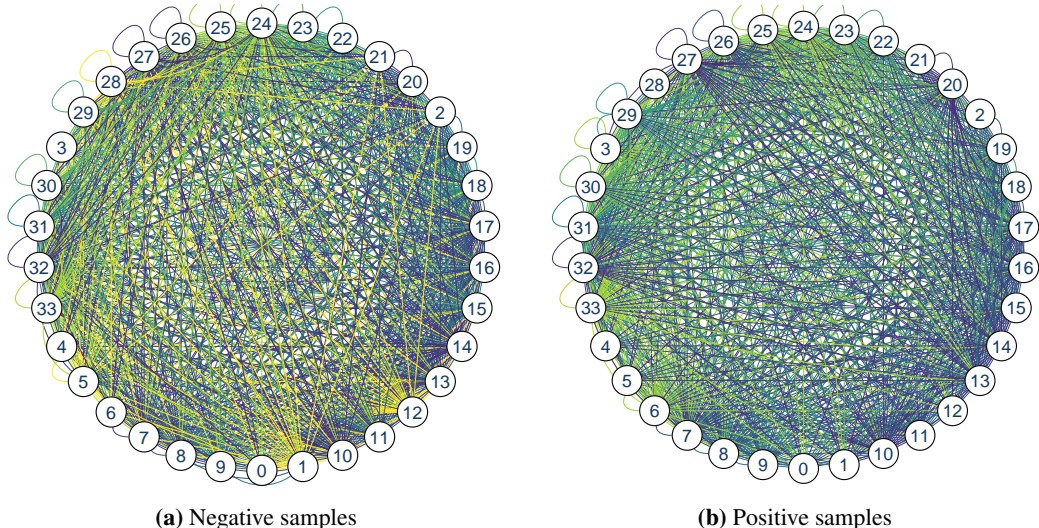

**(a)** Negative samples        **(b)** Positive samples

**Figure 4:** Learned structure for negative and positive samples (P19; Setting 1). The nodes numbered from 0 to 33 denote 34 sensors used in P19 (sensor names are listed in Appendix A.15). To make the visualized structures easier to understand, we use darker green to denote higher weight value and yellow to denote lower weight value. We can observe distinguishable patterns across two learned sensor dependency graphs, indicating RAINDROP is able to adaptively learn graph structures that are sensitive to the classification task. For example, we find that the nodes 1 (pulse oximetry), 5 (diastolic BP), and 12 (partial pressure of carbon dioxide from arterial blood) have lower weights in negative samples.

**Table 8:** Classification results when train and test samples originate from different groups (P19).

| Model | Generalizing to a new patient group | | | | | | | |
| --- | --- | --- | --- | --- | --- | --- | --- | --- |
| | Train: Young → Test: Old | | Train: Old → Test: Young | | Train: Male → Test: Female | | Train: Female → Test: Male | |
| | AUROC | AUPRC | AUROC | AUPRC | AUROC | AUPRC | AUROC | AUPRC |
| Transformer | $76.2 \pm 0.7$ | $30.5 \pm 4.8$ | $76.5 \pm 1.1$ | $33.7 \pm 5.7$ | $77.8 \pm 1.1$ | $26.0 \pm 6.2$ | $75.2 \pm 1.0$ | $30.3 \pm 5.5$ |
| Trans-mean | $80.6 \pm 1.4$ | $39.8 \pm 4.2$ | $78.4 \pm 1.1$ | $35.8 \pm 2.9$ | $80.2 \pm 1.7$ | $32.1 \pm 1.9$ | $76.4 \pm 0.8$ | $32.5 \pm 3.3$ |
| GRU-D | $76.5 \pm 1.7$ | $29.5 \pm 2.3$ | $79.6 \pm 1.7$ | $35.2 \pm 4.6$ | $78.5 \pm 1.6$ | $31.9 \pm 4.8$ | $76.3 \pm 2.5$ | $31.1 \pm 2.6$ |
| SeFT | $77.5 \pm 0.7$ | $26.6 \pm 1.2$ | $78.9 \pm 1.0$ | $32.7 \pm 2.7$ | $78.6 \pm 0.6$ | $31.1 \pm 1.2$ | $76.9 \pm 0.5$ | $26.4 \pm 1.1$ |
| mTAND | $79.0 \pm 0.8$ | $28.8 \pm 2.3$ | $79.4 \pm 0.6$ | $29.8 \pm 1.2$ | $78.0 \pm 0.9$ | $26.5 \pm 1.7$ | $78.9 \pm 1.2$ | $29.2 \pm 2.0$ |
| RAINDROP | $\mathbf{83.2 \pm 1.6}$ | $\mathbf{43.6 \pm 4.7}$ | $\mathbf{82.0 \pm 4.4}$ | $\mathbf{44.3 \pm 3.6}$ | $\mathbf{85.0 \pm 1.4}$ | $\mathbf{45.2 \pm 2.9}$ | $\mathbf{81.2 \pm 3.8}$ | $\mathbf{40.7 \pm 2.9}$ |

## A.14 FURTHER DETAILS ON ABLATION STUDY

We provide ablation study, taking PAM at Setting 1 as an example, in Table 7. In the setup of 'W/o sensor level concatenation', we take the average of all sensor embeddings (in stead of concatenating them together) to obtain sample embedding. Experimental results show that the full RAINDROP model achieves the best performance, indicating every component or designed structure is useful to the model. For example, we find that excluding inter-sensor attention weights $\alpha_{i,uv}^t$ will cause a decrease of 3.9% in accuracy while excluding edge weights $e_{i,uv}$ (i.e., dependency graphs) will drop the accuracy by 7.1%.

## A.15 VISUALIZATION OF INTER-SENSOR DEPENDENCY GRAPHS LEARNED BY RAINDROP

We visualize the learned inter-sensor dependencies (*i.e.*, $e_{i,uv}$ before the averaging operation in Eq. 3) on P19 in early sepsis prediction. The visualizations are implemented with Cytoscape (Shannon et al., 2003). The data shown are for testing set of P19 including 3,881 samples (3708 negative and 173 positive). As RAINDROP learns the specific graph for each sample, we take average of all positive samples and visualize it in Figure 4b; and visualize the average of all negative samples in Figure 4b. As we take average, the edges with weights smaller than 0.1 (means they rarely appear in graphs) are ignored. The averaged edge weights range from 0.1 to 1. We initialize all sample graphs as complete graph that has $1,156 = 34 \times 34$ edges, then prune out 50% of them in training phase, remaining 578 edges. The 34 nodes in figures denote 34 sensors measured in P19, as listed

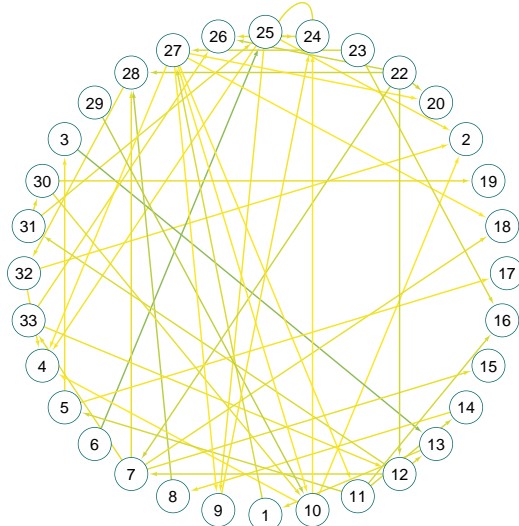

**Figure 5:** Differential structure of dependency graphs between positive and negative samples. The edges are directed. We select the top 50 edges with largest difference (in absolute value) between two patterns. The edges are colored by the divergences. The darker color denotes the connection is more crucial to classification task. Node 0 is not included in this figure as it is not connected with any sensor. We can infer that the heart rate is stable whether the patient will get sepsis or not. Moreover, we can see the edge from node 3 (systolic BP) to node 13 (Oxygen saturation from arterial blood) and the connection from node 6 (Respiration rate) to node 25 (Potassium) are informative for distinguishing sample classes.

https://physionet.org/content/challenge-2019/1.0.0/. We list the sensor names here: 0: HR; 1: O2Sat; 2: Temp; 3: SBP; 4: MAP; 5: DBP; 6: Resp; 7: EtCO2; 8: BaseExcess; 9: HCO3; 10: FiO2; 11: pH; 12: PaCO2; 13: SaO2; 14: AST; 15: BUN; 16: Alkalinephos; 17: Calcium; 18: Chloride; 19: Creatinine; 20: Bilirubin_direct; 21: Glucose; 22: Lactate; 23: Magnesium; 24: Phosphate; 25: Potassium; 26: Bilirubin_total; 27: TroponinI; 28: Hct; 29: Hgb; 30: PTT; 31: WBC; 32: Fibrinogen; 33: Platelets.

We also visualize the differential inter-sensor connections between the learned dependency graphs from patients who are likely to have sepsis and the graphs from patients who are unlikely to suffer from sepsis. Based on the aggregated graph structures of positive and negative samples, we calculate the divergence between two groups of patients and report the results in Figure 5. In detail, we sort edges by the absolute difference of edge weights across negative and positive samples. On top of the visualization of the 50 most distinctive edges, we can have a series of concrete insights. For example, the dependency between node 6 (Respiration rate) to node 25 (Potassium) is important to the early prediction of sepsis. Note these data-driven observations could be biased and still need confirmation and future analysis from healthcare professionals. The edges in both Figure 4 and Figure 5 are directed. The edge arrows might be difficult to recognize due to the small figure size. We will provide high-resolution figures to our public repository.

Furthermore, we statistically measure the similarities across samples within the same class and dissimilarities across samples from different classes. Specifically, for every sample, we calculate: 1) the average Euclidean distance between its dependency graph and the dependency graphs of all samples from the same class; 2) the average distance with all samples from the different classes. The P19 dataset has 38,803 samples including 1,623 positive samples and 37,180 negative samples. For a fair comparison, we randomly select 1,623 samples from the negative cohort, then mixed them with an equal number of positive samples to measure the averaged Euclidean distances intra- and inter-classes. We select the cohort for 5 independent times with replacement. We find that the distance $((8.6 \pm 1.7) \times 10^{-5})$ among dependency graphs of positive samples is smaller than the distance $((12.9 \pm 3.1) \times 10^{-5})$ across samples. The results show that the learned dependency graphs are similar within the same class and dissimilar across classes, which demonstrates RAINDROP can learn label-sensitive dependency graphs.

