# OpenReview forum: "Graph-Guided Network for Irregularly Sampled Multivariate Time Series"
_ICLR.cc/2022/Conference — ICLR 2022 Poster_

### Official Review · Reviewer_Vuyn · 2021-11-02

**Correctness:** 2
**Technical Novelty And Significance:** 3
**Empirical Novelty And Significance:** 3
**Recommendation:** 5
**Confidence:** 3

**Main Review:**

It is hard to understand the operation for the equation, and the figures in the paper are confusing, and the author does not provide any code to help us to check it.

For example,
1) The dot product in Figure 3 a is ambiguous.
Based on Appendix A.4, it seems the attention α and edge e are all real numbers.
Thus, in Figure 3 a, the dot product for α and e are multiple of the two real numbers without extra weight. However, the dot product of r||p and h is a dot product with matrix D between them. What’s more, the dot product between u and v is a dot product with two vectors between them.


What’s more, based on equation 3, e is calculated by α. Therefore, it seems there is no need to link u and e in Figure 3 a.

2) The form of β is confusing.
In Figure 3 b, the attention weight β seems to be a vector, but it is annotated as a real number instead of a vector symbol.  What’s more, the size of s in Equation 4 is not given; thus, it is hard to say the β is a vector or a real number.

3) The packed observation embedding in Figure 3 b is not mentioned in the paper.

 I tried to search the word “packed”. It is only shown in Figure 3 without explanation in any other places.


**Summary Of The Paper:**

The core idea of the paper is transforming multivariate time series data into a vector z and using z for classification.
The transformation has two steps. Firstly, each time record of each sensor is mapped into a  space of 20 dimensions. The first 4 dimensions are used for feature extraction, and the last 16 dimensions are used for positional encoding. Secondly, self-attention is used to transform the embedding into a vector z and use it for classification.


**Summary Of The Review:**

The presentation of the paper has big space to improve. The source codes are unavailable, and the main results cannot be confidently reproduced.

---

> ### Author Response · Authors · 2021-11-18
> **Response to Reviewer Vuyn**
>
> Thank you for your thoughtful commentary.
>
> We are sorry that the paper was not clear, and we will work to clarify it in the future, specifically focusing on flow, removing unnecessary abbreviations, and correcting any grammar mistakes.
>
> We respond to all your comments below. If you feel we have not sufficiently addressed your concerns to motivate increasing your score, we would love to hear from you further on what points of concern remain and how we can improve the work in your eyes. Thank you again!
>
> &nbsp;
> &nbsp;
>
> **Q about code availability**: The author does not provide any code to help us to check it.
>
> **Code availability**: We are afraid that is not quite right. In the initial submission, we provided a complete reference implementation of Raindrop together with code to reproduce experiments. See the anonymized link (footnote 1, Page 2) as well as our Reproducibility statement (Page 10), where the link is included in magenta: https://anonymous.4open.science/r/Raindrop). The link includes the code and datasets used, with reproduced implementations of all baselines. Further, we provided a detailed description of all data and scripts, configurations, along with examples of usage.
>
> &nbsp;
> &nbsp;
>
> **Q1.1**: The dot product in Fig. 3 a is ambiguous. Based on Appendix 4, it seems the attention α and edge e are all real numbers. Thus, in Fig. 3a, the dot product for α and e are multiple of the two real numbers without extra weight. However, the dot product of r||p and h is a dot product with matrix D between them. What’s more, the dot product between u and v is a dot product with two vectors between them.
>
> **A1.1**: We apologize for the confusing notation and will make it clearer. In the original Fig. 3a, the dot product symbol appeared three times.
> - The one between $\alpha^t_{i, uv}$ and $e_{i,uv}$ is a product of two real numbers.
> - The one on the horizontal arrow from $u$ to $v$ is a product between vector $\mathbf{h}^t_{i,u}$ and a real number (i.e., the product result of $\alpha^t_{i, uv}$ and $e_{i,uv}$).
> - The one between $\mathbf{h}^t_{i,u}$ and $\mathbf{r}_v || \mathbf{p}^t_i$ is a dot product between two vectors.
> -  The $\mathbf{D}$ is a weight matrix to adjust the dimension of $\mathbf{h}^t_{i,u}$, which is not important to the key idea and is not explicitly shown in Fig. 3a.
>
> We apologize for the ambiguous use of the dot product symbols. In response to the reviewer’s comment, we revised Fig. 3. We only keep the dot product symbol between vector $\mathbf{h}^t_{i,u}$ and vector $\mathbf{r}_v || \mathbf{p}^t_i$ and we remove other dot products to prevent confusion. All operations are clearly described in equations and text; moreover, we provide the implementation of Raindrop for readers to check.
>
> &nbsp;
> &nbsp;
>
> **Q1.2**: What’s more, based on Eq. 3, e is calculated by α. Therefore, it seems there is no need to link u and e in Fig. 3a.
>
> **A1.2**: In the revised Fig. 3a, we removed the link between node u and  $e_{i,uv}$. We are sorry for the typo.
>
> &nbsp;
> &nbsp;
>
> **Q2**: The form of β is confusing. In Fig. 3b, the attention weight β seems to be a vector, but it is annotated as a real number instead of a vector symbol. What’s more, the size of s in Eq. 4 is not given; thus, it is hard to say the β is a vector or a real number.
>
> **A2**: In the original submission, we mention on Page 6 (above Eq. 4) that: “The $\beta^t_{i,v}$ is the corresponding element of $\beta_{i,v}$”, indicating that $\beta_{i,v}$ is a vector and $\beta^t_{i,v}$  is an element of the vector. We are sorry for the typo that we did not bold the $\beta_{i,v}$.
>
> In the updated version, we made the following revisions:
>
> - We annotated all $\boldsymbol{\beta}_{i,v}$ using bold vector symbols (Fig. 3b; Section 3.4, Page 6).
>
> - We explicitly wrote “map …. to $T$-dimensional vector $\boldsymbol{\beta}_{i,v}$” in Section 3.4.
>
> - We clarified that $\mathbf{s}$ is a vector with $T$ elements that maps a $T \times T$-dimensional query-key matrix to a $T$-dimensional vector $\boldsymbol{\beta}_{i,v}$.
>
> &nbsp;
> &nbsp;
>
> **Q3**: The packed observation embedding in Fig. 3b is not mentioned in the paper.
> I tried to search for the word “packed”. It is only shown in Fig. 3 without explanation in any other places.
>
> **A3**: We appreciate your comment. In the original manuscript, we wrote that  "we pack the concatenated embeddings ..." in the second line of the second paragraph in Section 3.4 (Page 6). However, we acknowledge that is not easy to find and also is not a standard formulation. In the revised manuscript, we replaced the word ‘pack’ (both in Fig. 3 and the main text) with ‘stack’. The t-th row in the “Stacked observation embeddings” (Fig 3) denotes the product result of $[\mathbf{h}_{i, v}^t || \mathbf{p}^t_i ]$ and $\mathbf{W}$ in Eq(5).

---

> ### Author Response · Authors · 2021-11-25
> **Looking forward to your response**
>
> Dear reviewer,
>
> Thank you again for your thoughtful commentary. Following your suggestions, we included new results and clarified notation in the initial submission. We would love to hear your thoughts on our response. Please let us know if there is anything else we can do to address your comments. We would be very grateful if you considered increasing the score.

---

> ### Author Response · Authors · 2021-12-01
> **Looking forward to hearing from you**
>
> Dear reviewer,
>
> We sincerely appreciate your valuable comments on our work. In our previous response and the updated manuscript, we have tried our best to address the points raised in your review. Is there any unclear point that we can further clarify?
>
> Thank you again!

---

### Official Review · Reviewer_uubC · 2021-11-02

**Correctness:** 2
**Technical Novelty And Significance:** 3
**Empirical Novelty And Significance:** 2
**Recommendation:** 5
**Confidence:** 5

**Main Review:**

In this paper, the authors developed a GNN based methods for modeling irregular multivariate time series (MTS) and their classification. The proposed method draws an analogy between the ripple effects of raindrops and the propagation of intermittent observations of irregular MTS, and learns individual sensor embeddings through a neural message passing paradigm on the dependency graph between sensors. The goal is to learn MTS embeddings, which build upon sensor embeddings, which in turn build upon observation embeddings using temporal attention. During training, the embeddings and the dependency graphs were learned jointly, through the GNN related parameters. The dependency graph was inferred through observation-level attention across sensors. Thus, the whole model embeds the MTS in a hierarchical manner. In the experiments, the proposed method was compared with five related methods on irregular MTS, and was evaluated in several settings of missing patterns. The results indicate the effectiveness of the method. The paper provides a good perspective of applying GNN methods on MTS related tasks.

The concerns of the paper are summarized as below.
1. The introduction section claims "none of the existing methods on irregular MTS considered relational structures in representation learning ... irregular MTS" (the end of the second paragraph), which may not be true. The following several papers have taken into account of learning correlation parameters between sensors/variables in learning embeddings of irregular MTS.
	* Shukla and Marlin. "Interpolation-Prediction Networks for Irregularly Sampled Time Series." In ICLR. 2018. (cited in the paper)
	* Wu, et al. "Dynamic Gaussian Mixture based Deep Generative Model For Robust Forecasting on Sparse Multivariate Time Series." In AAAI. 2021.

	Although these papers didn't use GNN, but it is not rigorous to say relational structures have not been considered in the existing methods.

2. From the paper, it is unclear about why assuming inter-sensor dependency is specifically beneficial for dealing the irregularity of MTS, from an intuitive perspective. In other words, it is obscure on how important message passing is for dealing with the irregularity. Considering the dependency is useful for modeling the impacts between sensors, but what's more important to model irregular MTS, such as missing patterns and temporal impact decays (as discussed in many works in this area), are not well elaborated w.r.t. the proposed method.

3. In terms of the technique, it is unclear how could embedding and dependency graph be effectively learned in a joint manner. On one hand, the learning of embedding in Eq (2) depends on meaningful dependency graphs and pairwise attention weights. On the other hand, the learning of dependency graphs in Eq (3) depends on meaningful embeddings and attention weights. This forms a dilemma as neither of the embeddings nor the dependency graphs are available at the beginning. Moreover, the key component, inter-sensor attention in Eq (1) again depends on the embedding of the active sensor, but has little to do with the non-active sensors except for some variable parameters. From the current description, it is hard to understand how could the model training effectively learn both meaningful embeddings and dependency graphs.

4. The proposed method is sort of overcomplicated by introducing many learnable parameters, some are not well justified. For example, in Eq. (4), adding parameter s is for making the output a length-T vector, but what is the meaning of s, and its multiplication with the T-by-T self-attention matrix is unclear.

5. Since Eq (5) doesn't consider temporal order of MTS by using temporal attention, it is better to discuss if there is any impacts by neglecting the temporal structure during the aggregation.

6. The sample embedding is generated by concatenating all sensor embeddings. Since this may generate a long vector, will it limit the application of the proposed method when the number of sensors is large?

7. In the proposed method, many model parameters are shared across all samples for leveraging sample similarities, and there are few parameters for distinguishing different samples. Also, the pair-wise regularization in the loss function encourages similar dependency graphs among all samples. These designs may not be reasonable since not all samples are similar. It may be possible that samples form clusters or groups, some are similar while others are not. Encouraging similarity among all samples may be an unreasonable simplification.

8. In the experiment setting, it is better to summarize the missing data ratio in different datasets to highlight the irregularity in the these datasets. For the compared methods, many important ones were omitted, such as ODE based methods, the above two mentioned papers, GAN-based generative imputation methods (listed below), a memory-based method (listed below), and some GNN-based MTS method (listed below).
	* Luo, et al. "Multivariate time series imputation with generative adversarial networks." In NeurIPS. 2018.
	* Tang, et al. "Joint modeling of local and global temporal dynamics for multivariate time series forecasting with missing values." In AAAI. 2020.
	* Hu, et al. "Time-Series Event Prediction with Evolutionary State Graph." In WSDM. 2021.
	* Wu, et al. "Connecting the dots: Multivariate time series forecasting with graph neural networks." In KDD. 2020.

	As for the GNN-based MTS methods, although they don't directly model irregular MTS, it is feasible to first do statistics based imputation (mean imputation) and then apply them. This comparison helps understand how could the proposed method better address irregularity. If some of those methods have been compared on the same datasets in the same settings in other papers, it is better to report the numbers from there for a comprehensive understanding.

9. In the results of "Setting 4", it is unclear why the proposed method performs better than other methods when transferring from one group of samples to another, and what technical design helps the proposed method be effective in domain adaptation, which seems to be another topic different from modeling irregular MTS.

10. In the experiments, the visualization of the dependency graph cannot support the conclusion that the model learns dependencies for distinguishing classes. To this end, the visualization should demonstrate similarity of dependency graphs within the same class, and the dissimilarity between dependency graphs from different classes. Only observing dissimilarity across different classes provides few information. It is possible that the dependency graphs within the same class are different from each other, but the paper doesn't visualize nor discuss it. Also, considering samples with the same label as similar samples may not be reasonable in medical data because patients from the same class may have different data distribution due to different demographics, regions and population.

It is good to see the consideration of using visualization, as it facilitates the understanding of what have been really learned by the proposed model, especially considering the complexity of the model design. However, the current results make it difficult to judge whether the performance gains in Table 1, 2, 3 are from the correct learning of the attention and dependency graphs or other factors. Thus it is better to see a thorough visualization and discussion on inter-sensor attention, temporal attention, and the dependency graphs.

**Summary Of The Paper:**

This paper introduces a GNN based method for classifying irregular multivariate time series (MTS). The proposed method explicitly models the dependency between sensors, and propagates impacts of intermittent time series observations from sensors to related sensors for learning MTS embeddings. It jointly learns embeddings and dependency graphs through several attentive mechanisms. The paper provides a good perspective of applying GNN methods on MTS related tasks.

**Summary Of The Review:**

The paper is meaningful in exploring the new application of GNNs in irregular MTS related tasks. However, the paper is concerned w.r.t. its unclear description in some important concepts, the technical limitations, the lack of discussion and comparison of some related methods, and the non-comprehensive experiments.

---

> ### Author Response · Authors · 2021-11-18
> **Response to Reviewer uubC Part I**
>
> Thank you for your thoughtful commentary! They were very helpful, and we are glad you felt the paper provides a good perspective on applying GNNs to irregularly sampled time series.
>
> Below, we respond in detail to each of your concerns. If, after reading our responses, you do not feel we have sufficiently justified a higher score, please let us know where we can further improve our work or what concerns you still have so we can improve the work further. Thank you again!
>
> &nbsp;
> &nbsp;
>
>
> **Re Q1**: Thank you for your valuable comment. Our response to this has three parts.
>
> First, you are correct; the claim on “"none of the existing methods on irregular MTS considered relational structures in representation learning” was overstated. We update our claim in the text to be more specific (Section 1, Page 1-2).
>
> Second, IP-Net (Shukla & Marlin, 2018) and DGM$^2$-O (Wu et al., 2021) adopt imputation to interpolate the irregular time series against a set of reference points using kernel methods. The learned inter-sensor relations are static and do not consider sample-specific and time-specific characteristics (as evidenced by $\rho_{d d’}$ in Eq(5) in Shukla & Marlin 2018 and $\rho_{ij}$ in Eq(3) in Wu et al. 2021). In contrast, Raindrop avoids explicit interpolation based on kernels and instead uses dependency graphs. In particular, Raindrop models dynamic dependencies taking into account the varying influence of different sensors at different timestamps, an important characteristic of irregular time series. In addition, the above two papers assume all sensors are densely fully connected. In contrast, Raindrop learns sparse dependency graphs.
>
> Third, in response to the reviewer’s comment, we run a new experiment showing that Raindrop outperforms both IP-Net (Shukla & Marlin 2018) and DGM$^2$-O (Wu et al. 2021). In particular, Raindrop outperforms IP-Net by 14.2% and DGM$^2$-O by 8.4% in absolute accuracy on the PAM dataset. Results for Experimental Setting 1, PAM are in Appendix 11, Appendix Table 5, Page 18. We set hyperparameters of IP-Net and DGM$^2$-O methods following the authors’ recommendations. We will include additional results for other experimental settings in the camera-ready version. Further, we discuss the limitations of IP-Net and DGM$^2$-O in the revised section of related work (Section 2, Page 3).
>
>
> &nbsp;
> &nbsp;
>
>
> **Re Q2.1**: We did try to highlight the relevance of message passing for addressing the characteristics of irregular time series in our current submission, but apologize that point was not clear in the paper’s original presentation. In particular, observations in irregular time series are misaligned, meaning that at any given time point only a handful of sensors are observed and others have no measurements at all (i.e., are inactive). Previous studies (Wu et al., 2021; Li et al., 2020a; Zhang et al., 2019) found that inter-sensor correlations can capture rich information about how sensors behave.
>
> For this reason, we hypothesize that a) active sensor(s) can carry useful information about how currently inactive/unobserved sensors would behave, and that b) this information can be effectively leveraged to address the defining challenge of irregular time series, which is that observations are irregularly sampled with variable time between successive readouts and different subsets of sensors are observed at different time points (see our analogy with raindrops in Figure 1). Building on the hypothesis that sensor dependencies can alleviate the issue of irregular sampling, we model these dependencies with neural message passing. This is because neural message passing, and GNNs in particular, are the leading paradigm for modeling relational structure in current literature (observation that is not unique to us).
>
> To this end, we develop Raindrop to estimate the missing observations with neural message passing, based on the measured observations and the learned inter-sensor dependencies. We find that Raindrop consistently outperformed baselines in both established and hard, new experimental scenarios 1-4, which strongly validates our hypothesis. In the updated version, we included the intuition in Introduction (Section 1, Page 2).

---

> > ### Comment · Reviewer_uubC · 2021-11-26
> > **After reading response part 1**
> >
> > I would like to thank the authors for the extensive responses and revisions. Parts of the questions were answered, but there are some remaining ones that were not well addressed.
> >
> > ReQ1. The revised sentence is more reasonable than before. In the response, the authors also highlighted their method "learns sparse dependency graphs" but the other works "assume all sensors are densely fully connected". From the paper, the proposed method also learns densely connected dependency graph, and then pruning low weighted edges by manually setting a threshold. Therefore, in terms of computation, the proposed method also assumes the graph is fully connected. The computational complexity is similar to other works. Whether pruning graph with a threshold is better than keeping soft weighted edges or not seems have not been evaluated and is unclear in the studied applications. So it is not clear on what advantage is to highlight by mentioning this point.
> >
> > ReQ2.1. The hypothesis of using readings of active sensors to infer the readings of inactive sensors makes sense, perhaps with an assumption that there are sufficient active sensors (that are correlated with most of the inactive sensors) at a time step. If there are few active sensors (which is true in some applications, as demonstrated by the high missing ratio in ReQ8.1), the effectiveness of this method could be questionable. In this case, it may be important to also consider temporal dependency, that is to infer the reading of a sensor at a time step by considering its reading at the preceding time step. It seems not clear how could the proposed method leverage this information to avoid the case when there are few active sensors at some time steps.
> >
> > ReQ2.2. The point that the proposed method is complementary to modeling missing patterns is good, and the evaluation of adding mask matrices as input helps, but it is obscure whether "modeling missing patterns" includes learning task relevant missing patterns (such as GRU-D and IP-Net, which parameterize missing values).
> >
> > ReQ3.1. The description "the learning of graph dependency in Eq (3) only depends on pairwise attention weights" is not precise. Because, as mentioned in the response, the attention weights were computed from the embeddings of active sensors (which involve learnable parameters R) and another embedding r_{v} for the inactive sensors (with other parameters D and those for p), the learning of dependency graph really depends on the embeddings. It is not rigorous to describe the learning of graph dependency only depends on attention weights by neglecting Eq (1). Therefore, this question remains unclear. The authors used self-attention as an example, but it is hard to see how closely the proposed scenario connects to the self-attention learning. This makes some interpretable and qualitative results important (such as the visualization of graph and embeddings, on some small datasets is fine), which may help demonstrate that correct dependencies can be learned, but the paper lacks such results for justifying their design of methods.
> >
> > ReQ4. From the response, it is still obscure on why to first compute pairwise attention between embeddings at different time steps, and then aggregate them. It is not clear on how could aggregating pairwise relationships (at different time steps) generate a weight that represents the importance of a time step to the sensor. It is better to provide the intuition and the meaning of the steps, instead of elaborating the operations on matrices.
> >
> > ReQ6. From the response, it seems the datasets considered are not sufficiently comprehensive to cover cases with many sensors. In some applications, there could be thousands of sensors. Because the choice of the readout functions is sort of an empirical design, experiments with comprehensive datasets may be important. The authors may want to suggest what is the best readout function for the applications when there are many sensors, rather than providing several choices and letting the question be open, which may add overheads on evaluating different readout functions when using the proposed method.
> >
> > ReQ7. The response is ambiguous regarding the enforced similarity among all samples. To many methods, inputs naturally distinguish samples, but how does the enforced similarity, including the shared parameters and the pairwise regularization in the loss function, impact (to what extent) on the difference between samples is not clear. Perhaps by enforcing all dependency graphs to be similar, and sharing the parameters across all samples will largely reduce the inherent dissimilarity of some samples. The discussion in the revised Appendix is insufficient.

---

> > > ### Author Response · Authors · 2021-11-28
> > > **Response to the follow-up comments of Reviewer uubc (1/2)**
> > >
> > > We sincerely want to thank you for increasing the score. We will provide more details (like additional description of temporal attention weights) and improve the paper presentation (such as moving Setting 4 to the Appendix) in the final version following your comments.
> > >
> > > At this time in the rebuttal period, updates to the PDF are not allowed. For this reason, we respond below to your key follow-up comments. We would greatly appreciate your considering our additional response in your final score for this manuscript. Thank you again for your follow-up commentary!
> > >
> > > ---
> > >
> > > **ReQ2.1--When there are very few active sensors**
> > >
> > > Our response has two parts:
> > > - We did a new experiment to show that Raindrop can achieve good performance, outperforming a strong state-of-the-art baseline (i.e., Transformer-mean) even when very few sensors are active at each time step. As the PAM dataset is originally dense (sensors are fully observed), we manually adjusted the number of active sensors to investigate the influence of the number of active sensors on model performance. We randomly select N (N = 14, 10, 6, 2) out of 17 sensors as active sensors at each time step. We select different sensors across time steps; the selected sensors are fixed for all models for fair comparison. Sensors that are not selected are considered inactive and their observations are removed from the input. Results in the following table (Setting 1) show that the margin between Raindrop and Transformer-mean increased by 4% when the number of active sensors dropped from 14 to 2. It makes sense that performance drops with sensor sparsity increases. The important point, however, is that Raindrop has a bigger advantage over other MTS methods. The new experiment increases our confidence in Raindrop’s effectiveness when only a few sensors are active at a time. In the final version, we will include a full comparison (including weaker baselines) using this experimental setup.
> > >
> > > |     Models     |             #-active sensors                       |                    |                |                  |
> > > |------------------|:----------------------------------:|--------------|--------------|--------------|
> > > |            | 14 | 10 | 6 | 2 |
> > > | Raindrop | 92.3 +/- 1.2 | 89.6 +/- 1.8 | 86.2 +/- 2.1 | 76.5 +/- 2.7 |
> > > | Transformer-mean | 91.5 +/- 1.5 | 87.1 +/- 1.4 | 82.5 +/- 3.1 | 72.3+- 1.9 |
> > > | Margin | 0.8 | 2.9 | 4.5 | 5.8 |
> > >
> > >
> > > - We fully agree that modeling missing patterns can further help address the irregularity. We will include a discussion on this topic in the camera-ready version.
> > >
> > > &nbsp;
> > > &nbsp;
> > >
> > > **ReQ4--Intuition for the calculation of temporal attention weights**
> > >
> > > Thank you for this valuable suggestion. The intuition for temporal attention weights (Eq. 4) and $\mathbf{s}$ is as follows. Raindrop aggregates embeddings of observations recorded at different time steps into a single sensor embedding. To this end, we learn a weight for each observation and use the optimized weights to combine observation embeddings into a sensor embedding. But how to calculate these weights (i.e., temporal attention weights)? We start from widely used and standard self-attention that we modify as follows. The standard self-attention generates a `matrix of attention weights` (such that every element in the matrix denotes the relationship between two observations). However,  we only need a `vector of attention weights` (such that each element in the vector denotes the relationship between an observation and a sensor). For this reason, we use a trainable aggregator $\mathbf{s}$ that maps the `attention matrix` into an `attention vector`. The larger the weights in the attention vector the more important is a particular observation towards the sensor, and Raindrop will take more information from that observation when generating the sensor embedding. At last, using these attention weights, we combine observation embeddings into a single sensor embedding. We will explain this intuition behind Eq. 4 in the camera-ready version.
> > >
> > > Finally, conceptually, in this manuscript we recognize that irregular multivariate time series datasets are in general hierarchically organized (see Figure 2). That is, every sample has multiple sensors and, in turn, every sensor has many observations recorded in an irregular manner. To reason about samples (e.g., predicting outcome of the patient or leisure activity of the user), it is thus necessary to aggregate observations for each sensor and then aggregate sensors for each sample to generate sample embeddings. That is exactly what Raindrop does: Raindrop embeds individual observations considering inter-sensor dependencies, aggregates them into sensor embeddings using temporal attention, and finally integrates sensor embeddings into sample embeddings.
> > >
> > > &nbsp;

---

> > > ### Author Response · Authors · 2021-11-28
> > > **Response to the follow-up comments of Reviewer uubc (2/2)**
> > >
> > > **ReQ7--Influence of the enforced similarity of dependency graphs**
> > >
> > > That is a great point, thank you. To address it, we did a new experiment to evaluate the influence of dependency graphs on dissimilarity between samples. We investigate dissimilarity in two ways: 1) by evaluating input observations and 2) by evaluating learned sample embeddings. In particular, we measure the average Euclidean distance between all sample pairs. Results are in the following table, where *with  $\mathcal{L}_r$* indicates that Raindrop is encouraged to learn similar dependency graphs for all samples and *w/o $\mathcal{L}_r$* indicates that no sample regularization is used in the loss function. Starting from the initial distance between input observations to distance between optimized sample embeddings, we see that distances among samples have increased. This makes sense and is the desired effect because optimized embeddings should have greater discriminative power than input data so that they can work better with downstream classifiers. Importantly, we find that the distance between sample embeddings does not depend considerably on the inclusion of the regularizer. We conclude that the enforced similarity among all samples does not largely affect the inherent dis/similarity between samples.
> > >
> > > | Datasets | Regularization | Eucl. dist. input observations | Eucl. dist. sample embeddings |
> > > |----------|----------------------|--------------------|-------------------|
> > > | P19 | with $\mathcal{L}_r$ | 1.8*10$^{-4}$ | 2.3*10$^{-4}$ |
> > > | P19 | w/o $\mathcal{L}_r$ | 1.8*10$^{-4}$ | 2.2*10$^{-4}$ |
> > > | PAM | with $\mathcal{L}_r$ | 1.1*10$^{-3}$ | 1.9*10$^{-3}$ |
> > > | PAM | w/o $\mathcal{L}_r$ | 1.1*10$^{-3}$ | 2.1*10$^{-3}$ |
> > >
> > > We will add details to Appendix 4 in the final version.
> > >
> > > &nbsp;
> > >
> > > **ReQ10.1--Clarify the calculation of Euclidean distances; use correctly predicted samples for distance calculation**
> > >
> > > Thanks for recognizing the value of our quantitative evaluations.
> > > -Yes, the Euclidean distance between dependency graphs quantifies the difference between adjacency matrices of the graphs (one matrix/graph is learned for each sample). For every pair of positive samples, we calculate the distance between their adjacency matrices and regard the average distance as `intra-positive class similarity.` For every pair of samples with different labels (i.e., a pair consisting of one positive and one negative sample), we calculate the Euclidean distance and take the average distance to obtain `inter-class dissimilarity.` As reported in initial response to Q10.1, results show that intra-class distance is smaller than inter-class distance, which is the desired behavior.
> > > -To address your comment, we updated distance calculation and limited calculations to only correctly predicted test samples instead of all samples. For P19 under Setting 1, there are 112 positive test samples and 2590 negative test samples that are correctly classified. To compare the distance between cohorts of similar size, we randomly selected 112 samples from the negative set, then mixed them with an equal number of positive samples to measure average Euclidean distance for intra- and inter-classes. We independently repeated this process 5 times and we report average results here. The intra-class distance (7.2 \pm 1.2) * 10$^{-5}$ between positive samples is smaller than inter-class distance (9.8 \pm 1.3)* 10$^{-5}$ between positive and negative samples. We find that confidence intervals do not overlap in this setting. Thank you for the suggestions!
> > >
> > > We will include evaluation on other datasets in the camera-ready version.
> > >
> > > &nbsp;
> > >
> > > **ReQ10.3-The number of hyperparameters**
> > >
> > > Raindrop has some general hyperparameters (e.g., learning rate, batch size, hidden layer dimension, the number of layers) that exist in essentially all deep models (including existing MTS methods) and are not unique to Raindrop. Beyond those, Raindrop has only six unique hyperparameters: dimension $d_a$ of time representation, dimension $d_a$ of static information, dimension $d_r$ of sensor-specific vector $\mathbf{r}_v$, dimension $d_h$ of observation embedding, regularization coefficient $\lambda$ in the loss, and pruning rate $K$.
> > >
> > > We thus conclude that the number of hyperparameters in Raindrop is comparable to other state-of-the-art MTS methods like SeFT and mTAND. In particular, SeFT has four unique hyperparameters: dimension of time representation, dimension of static information, and the output size of $\theta$ and $phi$ functions. mTAND also has four unique hyperparameters: dimension of time representation, number of reference points, dimension of GRU’s output, and embedding size of the input. In response to your comment, we will provide details on hyperparameter selection in the public code repository.
> > >
> > > ---
> > >
> > > We would greatly appreciate it if you can consider our additional response in your final recommendation and the score for this manuscript. Thank you again for your constructive commentary!

---

> > ### Comment · Reviewer_uubC · 2021-11-26
> > **After reading response part 2**
> >
> > ReQ8.2. I sincerely appreciate the added experiments, which help demonstrate the effectiveness of the proposed method to some extent. I also understand in the previous manuscript, the authors described mTAND outperformed many other methods in its reported results. As mentioned in my previous question 8, the authors may also want to clarify whether the reported results by mTAND are on the same datasets and in the same settings as in the manuscript. If so, those results may be included in the manuscript for a comprehensive understanding. If not, perhaps it is hard to say whether different choices of datasets and settings can provide the same conclusion on method comparison (and it may be worth description on why not to use the same datasets as mTAND). As for the added results, it may be better to report full results on all datasets in the experiments and move them to section 4.1 from the Appendix.
> >
> > ReQ9. The response seems not convincing. Sample-specific predictions of dependency graphs means the method is inductive. It may not mean the method has the power of domain adaptation. If the authors think their method is not designed for solving the problem of domain adaptation, it may be better to remove setting 4 or put it in the Appendix to avoid the confusion or distraction. Adding setting 4 and describing the good performance without clear reasons in the manuscript may lead to many questions. Instead, the experiments are better to be designed to support the focus of the work.
> >
> > As for the added experiment, it may not validate that removing dependency graphs reduced the domain adaptation ability of the proposed method. It is likely that removing dependency graphs also reduces the performance in Setting 1, which means removing dependency graphs may reduce the performance of the proposed method in general, rather than specific to setting 4. Thus the domain adaptation ability of the proposed method may have not been justified.
> >
> > ReQ10.1. It is a good perspective to quantitatively evaluate the similarity and dissimilarity of the learned dependency graphs in different classes. Perhaps it is better to describe how the Euclidean distances on graphs were calculated. Are they calculated on the difference between adjacency matrices? Also, it may be more reasonable to compare the distances using correctly predicted testing samples than using all (or random) training samples. This may help see if the difference of distances correlates with the predictions. The evaluation is better to be performed on all datasets in the experiments than using one dataset. From the reported distances, it seems the confidence intervals have overlap, it is not clear whether the difference of the distances is significant or not.
> >
> > ReQ10.3. It seems Fig 5 convey few information because the conclusion on the importance of the learned edges in Appendix 15 was not effectively validated. The ablation study helps clarify the effectiveness of including attention weights and dependency graphs, perhaps in a sense that those components added model parameters. It may not explicitly indicate whether the performance gains are from the learning of correct attention weights and dependency graphs, which should be interpretable. Current visualization results do not provide convincing evidences that the learned dependency graphs are meaningful, and encode class-wise patterns that interprets the performance gains.
> >
> > Another possible limitation of the proposed method is that it appears to have many hyperparameters, because it introduced many learnable parameters which require setup of dimensions. Setting all of the hyperparameters may make applying the method not easy. The impacts of some other hyperparameters, such as the ratio of edge pruning, K, and the trade-off parameter, lambda, in the loss function, lack evaluation and discussion (on how to choose) in the manuscript.
> >
> > Also, I agree with another reviewer that the presentation of the paper remains to be improved. Fig 3 and some equations are not easy to follow because of the complexity of the method and some design choices that were not justified very well. In the meantime, I appreciate the author responses and the revisions that addressed some of the questions. Therefore, weighting up the pros and cons of the work, I increased the score from 3 to 5.

---

> ### Author Response · Authors · 2021-11-18
> **Response to Reviewer uubC Part II**
>
> **Re Q2.2**: We agree that missing patterns (such as masking and variable time intervals between adjacent readouts) and temporal decays are important. Our response has four parts:
> - In this work, we propose an alternative perspective on modeling irregular time series. This perspective is both beneficial (see benchmarking results) and necessary (see ablation study) in the sense that it is complementary to explicit imputation of missing patterns and temporal decay. The key point in this paper is that Raindrop is beneficial strategy for irregularity, which is neither in conflict nor overlapping with the explicit modeling of missing patterns.
>
> - Moreover, a number of recent state-of-the-art methods do not explicitly consider missing patterns and temporal decay as inputs. Yet, these methods were shown to achieve state-of-the-art performance at the time of their publication. For example, mTAND (ICLR’21) models the irregularity using a continuous-time embedding mechanism, and SeFT (ICML’20) represents irregular MTS with set functions.
>
> - Further, temporal decay and missing patterns are closely tied to sequential processing; however, recent studies shown that self-attention (such as mTAND, SeFT, and Transformer) perform better than RNNs. Thus, we use temporal self-attention instead of sequential processing. For example, we empirically shown that Raindrop, which relies on temporal attention, outperforms GRU-D, which is a strong model that explicitly leverages missing patterns and temporal decay.
>
> - We think that adding experiments on this would be insightful. For this reason and to empirically address the reviewer’s comment, we demonstrate that considering missing patterns can help further improve Raindrop’s performance. We conducted an extensive experiment (Appendix 9, Page 17) which shows that concatenating the masking matrix with observations as inputs can slightly boost the model’s performance by 1.2% (P19; Setting 1). Importantly, these results do not conflict with the key contributions of this paper at all for reasons mentioned in the above three points. We thank you for this comment, and we will work hard to clarify so this message is better conveyed.
>
>
> &nbsp;
> &nbsp;
>
>
> **Re Q3.1**: Our response has two parts.
>
> - We first clarify that the learning of graph dependency in Eq (3) only depends on pairwise attention weights $\alpha^t_{i,uv}$ but not directly on embeddings. Although the attention weight $\alpha^t_{i,uv}$ is learned based on embedding $\mathbf{h}^t_{i,u}$ in Eq(1), the $\mathbf{h}^t_{i,u}$ (embedding of active sensor; generated from the input observation $x^t_{i, u}$) is not the same with the embedding $\mathbf{h}^t_{i,v}$ in Eq(2) (embedding of non-active sensor; unknown, learned in Eq(2)). Thus, we don’t need to be concerned with the dilemma.
>
> - Moreover, we agree that embeddings and dependency graphs are both automatically learned by the model based on the input observations. However, this is not unique to Raindrop but broadly exists in attention mechanism-related publications. Previous studies (such as [r1-r4] below) have shown that deep neural networks can effectively learn two meaningful variables even if both of them are not available at the beginning.  For example, in the classic Transformer architecture, the self-attention weights and latent embeddings are not given at the beginning and are learned by the model.
>
> [r1] Al-Rfou, R., et al., 2019. Character-level language modeling with deeper self-attention. AAAI.
>
> [r2] Yun, S., et al., 2019. Graph transformer networks. NeurIPS.
>
> [r3] Hu, Z., et al., 2020. Heterogeneous graph transformer. WWW.
>
> [r4] Vaswani, A., et al., 2017. Attention is all you need. NeurIPS.
>
>
> &nbsp;
> &nbsp;
>
>
> **Re Q3.2**: This work studies irregularly sampled time series, meaning that observations for inactive sensors are missing and so it would not be appropriate to consider embeddings of inactive sensors in Eq. (1). To address this characteristic of irregularly sampled time series and to model inter-sensor attention weights, our solution is to take a well-designed variable $\mathbf{r}_v$, which is specific to an inactive sensor $v$, as a proxy for the inactive sensor’s information. We believe this solution is reasonable and well-motivated by the data characteristics. This confidence is further strengthened by excellent performance of Raindrop in the ablation study (the F1-score decreased steeply by 7% when removing variables $\mathbf{r}_v$ specific to inactive sensors).

---

> ### Author Response · Authors · 2021-11-18
> **Response to Reviewer uubC Part III**
>
> **Re Q4**: We apologize for any ambiguity in our theoretical specifications and the confusing notation that we will make clearer. In particular, expression in Eq. (4) describes how we learn the temporal attention weights vector $\boldsymbol{\beta}_{i,v}$ for sensor $v$, following the self-attention formalism. The standard dot-product self-attention generates a matrix of weights that have the attention weights for each pair of observations. The intuition of the learnable $\mathbf{s}$ is to measure the importance of observation at a specific time step for the whole sensor embedding, based on the timewise relationships between observations.
>
> Next, we give a concrete example to specifically describe the meaning of $\mathbf{s}$. Each row, j, of the self-attention matrix captures relationships of observation embedding at timestamp $t_j$ to all observation embeddings at $\{t_k: k=1, \cdots,T}$. Then, using the learnable weight vector $\mathbf{s}$, these correlations are aggregated across time to obtain temporal importance weight $\beta^{t_j}_{i, v}$.
>
> The multiplication is a matrix product between the self-attention matrix (with shape $T \times T$) and $\mathbf{s}$ (with shape $T \times 1$), which generates the result with shape $T \times 1$. In the result, every element $\beta^{t_j}_{i, v}$ represents the importance of the corresponding observation to the whole sensor embedding.
>
> In response to your comment, we added Appendix 2 (Page 15) in the updated version to describe parameter $\mathbf{s}$ in detail. We also explicitly report the dimensionality of $\mathbf{s} \in \mathbb{R}^{T \times 1} $ and clarify that the multiplication between $\mathbf{s}$ and the T-by-T self-attention matrix is a matrix product (Section 3.4, Page 6).
>
>
> &nbsp;
> &nbsp;
>
>
> **Re Q5**: We think there may be a fundamental misunderstanding here, for which we apologize. While we will clarify the text on this point, we would like to highlight here two things.
>
> We believe this question is caused by misunderstanding because Raindrop does consider temporal ordering of observations. In particular, although we do not process signals in sequence, we take into account temporal structure by encoding time representation $\mathbf{p}^t_i$ as input of Eq. (5). In particular, we encode the timestamp (a continuous value, not discrete position number) into a time representation (Hornet al., 2020)  which can provide rich temporal information in the generation of sensor embedding (Appendix 1, Page 16).
>
> In response to your comment, we conducted an experiment comparing the effectiveness of temporal attention to LSTM. In particular, we replace the temporal attention in sensor embedding generation (Eq. (4-5)) in Raindrop by the LSTM layer which processes all observation embeddings in order. We use zero padding to convert the irregular observations into fixed-length time series so that the data can be fed into LSTM architecture. We regard the last output of LSTM as generated sensor embedding. The number of LSTM cells equals the dimension of observation embedding. All the model structures are identical except the difference between temporal attention and LSTM. We keep all experimental settings (P19; Setting 1) and hyperparameter selections the same. Results show that temporal self-attention outperforms LSTM by 1.8% (AUROC) and saves 49% of training time. This might be due to attention allowing parallel computation and also alleviating performance degradation otherwise caused by long-term dependencies (Ganesh et al., 2021; Vaswani et al., 2017). We include new experiments in Appendix 10 (Page 18) of the updated version.
>
>
> &nbsp;
> &nbsp;
>
>
> **Re Q6**: That is a great point to which we respond in two parts.
> - As indicated in the initial submission (the second and third sentences in Section 3.5), Raindrop can be used with any readout function (e.g., Errica et al., 2021, ICML) that aggregates sensor embeddings into sample embeddings. This naturally solves the question of long vectors. In this work, we use *concatenation* through all experiments as the number of sensors in P12, P19, and PAM datasets is rather small.
> - Concatenating sensor embedding can give a lengthy sample embedding on datasets with a large number of sensors, yet this question can be easily addressed. For instance, we can reduce the dimensionality of sample embeddings by adding an additional neural network layer (such as a fully-connected layer) following the concatenation. We also did a new experiment showing that concatenation works better than averaging (we see a boost in the AUROC score by 0.6% using concatenation instead of averaging for generating sample embeddings; P19; Setting 1). In response to your comment, we include the above discussion and experiments in the updated version (Appendix 3, Page 15).

---

> ### Author Response · Authors · 2021-11-18
> **Response to Reviewer uubC Part IV**
>
> **Re Q7**: Thank you very much for your valuable perspective here! Our response has two parts.
>
> First, distribution shifts over domains (such as domain generalization and subpopulation shifts mentioned in your comment), while an incredibly important research field (synthesized by benchmarks like WILDS, https://wilds.stanford.edu), are neither the focus nor the motivation for this study. Our study focuses on the question of irregularity rather than the question of distribution shifts in time series. To this end, in our experiments, we first rigorously benchmark Raindrop using a standard evaluating setup (Setting 1, which is classification of irregular time series). This is the only setup that most existing methods consider (e.g., mTAND, GRU-D) and we wanted to make sure our comparisons are fair. In order to provide a more rigorous assessment of Raindrop’s performance, we also consider more challenging setups in our experiments (i.e., Settings 2-4) when the dataset is evaluated in a non-standard manner and the split is informed by a select data attribute. Our results on Setting 1 are consistent with those on Settings 2-4. Results on harder Settings 2-4 show that Raindrop can perform comparably better than baselines. Results across these diverse settings increase our confidence that Raindrop is quite flexible and widely applicable. We did try to highlight these points in our submission, but apologize if we ever seemed to be implying that Raindrop is solving the problem of subpopulation shift or modeling the latent clustering structure. We agree that would be an unreasonable simplification.
>
> Second, different samples are naturally distinguished by their inputs. In this work, to model sample-specific characteristics, Raindrop learns, for each sample, a set of distinct inter-sensor attention weights, a dependency graph, and temporal self-attention weights. It is precisely because so many weights are not shared across samples that we emphasized in the text on parameters that are shared across samples. In response to your comment, we also discuss similarities between samples in the newly added Appendix 4 (Page 15).
>
>
> &nbsp;
> &nbsp;
>
>
> **Re Q8.1**: That is a great suggestion. The missing ratios for P19, P12, and PAM datasets are 94.9%, 88.4%, and 60%, respectively. In the revised version, we included a table (Appendix 5, Appendix Table 4, Page 16) showing dataset statistics.
>
> | Datasets | \#-samples | \#-sensors | \#-timestamps | \#-classes | Missing ratio (%) |
> |----------|------------|------------|---------------|------------|--------------------|
> | P19      |     38,803 |         34 |            60 |          2 |               94.9 |
> | P12      |     11,988 |         36 |           215 |          2 |               88.4 |
> | PAM      |      5,333 |         17 |           600 |          8 |              60 |

---

> ### Author Response · Authors · 2021-11-18
> **Response to Reviewer uubC Part V**
>
> **Re Q8.2**: We thank the reviewer for bringing up these papers. Our response has three parts, and we consider all these papers and more.
>
> - In the initial submission, we benchmarked Raindrop against the most competitive and most relevant baselines. For example, mTAND (Shukla and Marlin 2021) is the latest work from the authors of IP-Nets (Shukla and Marlin 2018), so we compared Raindrop with mTAND instead of IP-Nets. We did not consider EvoNet (Hu, et al. 2021) and MTGNN (Wu, et al. 2020) because they are not designed for irregular time series. As indicated in our initial related work, our study focuses on time series classification (i.e., predicting a label for a given sample), which is a distinct problem different from forecasting (i.e., anticipating future observations based on historical observations, like DGM$^2$ and Tang et al. 2020). Because of that, Raindrop is primarily benchmarked against state-of-the-art classification methods.
>
> - As we explicitly state in the initial manuscript (Section 4, paragraph on baselines) that: “The mTAND (Shukla & Marlin, 2021) method has been shown to outperform numerous recurrent models including RNN-Impute (Che et al., 2018), RNN-Simple, Phased-LSTM (Neil et al., 2016), and IP-Nets (Shukla & Marlin, 2018), along with ODE-based models such as LATENT-ODE and ODE-RNN (Chen et al., 2018). For this reason, we compare with mTAND and do not report comparisons with those techniques in this paper.“ Experiments show that Raindrop consistently outperforms mTAND, and so, by extension, this indicates that Raindrop would also perform favorably against baselines considered in the mTAND paper.
>
> - In response to your comment, we performed new experiments comparing Raindrop with five methods: ODE-RNN (Chen et al., 2018), IP-Net (Shukla and Marlin 2018), DGM$^2$-O (Wu, et al. 2021), EvoNet (Hu, et al. 2021), and MTGNN (Wu, et al. 2020). Note that Luo et al. 2018 and Tang, et al. 2020 don’t provide open-access code and so we cannot faithfully reimplement these two methods in this short response period. When implementing the above five methods, we follow the authors’ recommendations on hyperparameters. For methods, which cannot deal with irregular data (i.e., EvoNet and MTGNN), we first impute the missing data using mean imputation and then feed data into the model. For forecasting models (i.e., MTGNN) which are strictly not comparable with the proposed classification model, we formulate the task as a single-step forecasting, concatenate the learned representations from all sensors and feed into a fully-connected layer (work as classifier) to make classification, and use cross-entropy to quantify the loss.
> The following table shows results for the PAM dataset and Setting 1. Results show that Raindrop outperforms the strongest baseline in accuracy by 4.7%. The added experiments increase our confidence in the effectiveness of Raindrop.
>
> We will include additional comparison experiments on other settings and datasets in the camera-ready version.
>
> | Models           | Accuracy     | Precision    | Recall       | F1 score     |
> |------------------|--------------|--------------|--------------|--------------|
> | ODE-RNN          | 79.8 +/- 3.6 | 81.2 +/- 2.7 | 81.9 +/- 2.4 | 81.7 +/- 2.6 |
> | IP-Net           | 74.3 +/- 3.8 | 75.6 +/- 2.1 | 77.9 +/- 2.2 | 76.6 +/- 2.8 |
> | DGM^2-O          | 80.1 +/- 2.3 | 80.6 +/- 1.9 | 81.2 +/- 1.8 | 80.9 +/- 2.1 |
> | EvoNet           | 84.5 +/- 2.7 | 84.8 +/- 1.8 | 85.6 +/- 3.1 | 85.3 +/ 2.3  |
> | MTGNN            | 83.4 +/- 1.9 | 85.2 +/- 1.7 | 86.1 +/- 1.9 | 85.9 +/- 2.4 |
> | Raindrop         | 88.5 +/- 1.5 | 89.9 +/- 1.5 | 89.9 +/- 0.6 | 89.8 +/- 1.0 |
>
> Results for five other baselines (Transformer, Transformer-mean, GRU-D, SeFT, and mTAND) on this setting are in Table 1 of our original submission.

---

> ### Author Response · Authors · 2021-11-18
> **Response to Reviewer uubC Part VI**
>
> **Re Q9**: In Setting 4, we wanted to test Raindrop under a challenging scenario. We apologize if we ever seemed to be implying that Raindrop is solving the problem of domain adaptation. We fully agree that this is a topic different from modeling irregular MTS. However, it is reasonable that Raindrop performs well in Setting 4. One reason is that Raindrop’s estimates inter-sensor weights and optimizes dependency graphs in a manner that is sample-specific. Thus, Raindrop has the power, to some extent, to adaptively estimate inter-sensor dependencies that can generalize to test samples. This means that Raindrop can generalize to new samples but not necessarily new groups: this results in a good performance although Raindrop is not designed for domain adaptation.
>
> The above-mentioned reasoning can be validated empirically. To do that, we add a new experiment, where we remove inter-sensor dependencies, i.e., set all sensors to be isolated nodes in Raindrop’s dependency graphs (i.e., set all $\alpha^t_{i, uv}$ and $e_{i, uv}$ to 0). We then test this reduced version of Raindrop on Setting 4. Results show that performance drops by 3.6% (AUROC) when excluding dependency graphs and message passing in Raindrop. We included discussion and this experiment in the revised manuscript version (Appendix 13, Appendix Table 8, Page 19). We did try to highlight these points in our current submission, but apologize if we ever seemed to be implying that Raindrop is solving distribution shifts over domains.

---

> ### Author Response · Authors · 2021-11-18
> **Response to Reviewer uubC Part VII**
>
> **Re Q10.1**: Thank you very much for the valuable comment. As visualizing thousands of separate dependency graphs is extremely difficult in practice, in the revised version, we statistically measured the similarities across samples within the same class and dissimilarities across samples from different classes. Specifically, for every sample, we calculated the average Euclidean distance between its dependency graph and the dependency graphs of all samples from the same class, and also measured the average distance with all samples from the different classes. The P19 dataset has 38,803 samples including 1,623 positive samples and 37,180 negative samples. For fair comparison, we randomly selected 1,623 samples from the negative cohort, then mixed them with an equal number of positive samples to measure the averaged Euclidean distances intra- and inter-classes. We selected the cohort for 5 independent times with replacement and reported the average results. We found that the distances ( (8.6 +/- 1.7)*10$^{-5}$ ) among dependency graphs of positive samples is smaller than the distance ( (12.9 +/- 3.1)*10$^{-5}$) across samples. The results show that the learned dependency graphs are similar within the same class and dissimilar across classes, which demonstrates Raindrop can learn label-sensitive dependency graphs. We have reported the results and discussion in Appendix 15 (Page 20).
>
>
> &nbsp;
> &nbsp;
>
>
> **Re Q10.2**: Our response has two parts.
>
> - Dependency graphs are learned automatically in a supervised manner such that errors are calculated and back-propagated based on class labels. Therefore, the optimized dependency graphs should be sensitive to the class label, and samples with the same label are more likely to have similar graph structure. To address your point, we updated the sentence “We investigate whether similar samples (i.e., with same label) have similar graph structures ...”  to “We investigate whether samples with the same label have similar graph structure” (Section 4.2, Page 9) to be more precise in the revised version.
>
> - Nota that domain shift caused by distribution shifts over medical data domain is not the problem addressed by Raindrop. Domain adaptation or domain transfer are not the goal of this work, and we hope this is sufficiently clear from our responses in Re Q7 and Re Q9.
>
>
> &nbsp;
> &nbsp;
>
>
> **Re Q10.3**: Thank you for recognizing our efforts on visualization. We appreciate your comments. Our response has two parts.
>
> - In the revised manuscript, we provide an extended analysis of differential dependency graphs. We visualized the inter-sensor connections between the learned dependency graphs from patients who are likely to have sepsis and the graphs from patients who are unlikely to suffer from sepsis. We calculated the divergence between two groups of patients and reported the results in a newly added Appendix Fig. 5. In detail, we sorted edges by the absolute difference of edge weights across samples with different labels. More details on the new figure and discussions are in Appendix 15, Appendix Fig. 5, Page 20-21. We will include high-resolution visualizations in our public Github repository. Further, we will present more thorough visualizations and discussions in the camera-ready version.
>
> - We provide an extended ablation study to show that gains in performance are due to effective learning of inter-sensor attention weights and dependency graphs. For example, in Appendix 14, Appendix Table 9 (PAM; Setting 1), we found that excluding inter-sensor attention weights decrease performance by 3.9% in accuracy and that excluding edge weights (i.e., dependency graphs) decreases performance by 7.1%.

---

> ### Author Response · Authors · 2021-11-23
> **Looking forward to your feedback**
>
> Dear reviewer,
>
> Thank you again for your valuable suggestions on improving this manuscript. To address your concerns, we performed a number of additional experiments, included new tables and figures in the paper, and revised the text.
>
> We would love to know what you think about our response and if there is anything else we can do to improve the paper. We would greatly appreciate your considering increasing the score. Many thanks!

---

### Official Review · Reviewer_cdpu · 2021-11-02

**Correctness:** 4
**Technical Novelty And Significance:** 3
**Empirical Novelty And Significance:** 4
**Recommendation:** 8
**Confidence:** 4

**Main Review:**

I think the model design is reasonable and the results are impressive. It is interesting to see that the dependency between different sensors really helps the predictions in Setting 2,3,4, which will be very useful for many applications. Only thing is that the learned graph (Figure 4) are very dense and it will be interesting to see what happens by making it sparse.

**Summary Of The Paper:**

This manuscript describes a method that models multiple time series data (sensors) across different individuals (samples). A graph is used to model the dependency between the sensors, such that the data of one sensor could be used to infer another, which is the key point of this ms. How one sensor affects another is modelled as a message passing problem on the graph. At each time point, an observation embedding is generated, either from a sensor or its neighbours in the graph. Observation embeddings of a sensor is then aggregated into a fixed-length sensor embedding, using temporal attention. These sensor embeddings are concatenated into a sample embedding for an individual for downstream analysis such as classification.

**Summary Of The Review:**

I recommend this ms to be accepted.

---

> ### Author Response · Authors · 2021-11-18
> **Response to Reviewer cdpu**
>
> Thank you for your helpful feedback! We appreciate that you felt our method is reasonable and that you recognize the results are impressive and very useful for many applications.
>
> **Q1**: I think the model design is reasonable and the results are impressive. It is interesting to see that the dependency between different sensors really helps the predictions in Setting 2,3,4, which will be very useful for many applications. Only thing is that the learned graphs (Figure 4) are very dense and it will be interesting to see what happens by making it sparse.
>
> **A1**: Thank you for this valuable comment. To address your comment, we reduced the number of edges to make Fig. 4 sparser. In particular, we now visualize the top 50 edges with the largest difference between class labels. The new figure is included as Appendix Fig. 5 (Page 21) in the revised version. More details and discussions are provided in Appendix 15 (Page 20-21). We will present more thorough visualizations and discussions in the camera-ready version.

---

### Author Response · Authors · 2021-11-18
**General Response**

We thank all reviewers for their valuable comments. We are pleased the reviewers felt our work is a reasonable and interesting approach that provides a good perspective on how to apply GNN methods on MTS tasks, that it is useful for many applications, and that results are impressive.

We address all reviewer’s comments directly in individual replies. In response to reviewers’ comments, we added 7 appendices, conducted 6 new experiments, 3 new tables, and 1 new figure. In addition, we revised the manuscript to improve notation and include additional clarifications. We uploaded the revised manuscript to Open Review and highlighted key revisions in blue.

---

### Decision · Program_Chairs · 2022-01-20

**Decision:**

Accept (Poster)

**Comment:**

The authors introduce a GNN based method for classifying irregular multivariate timeseries.
They represent the dependencies among sensors using a graph structure and deploy message passing to
model the effect of a sensor on another)s=. The approach jointly learns embeddings and the dependency graph.

The manuscript gathered a clear accept (8) and two marginal below the threshold scores (5).
I want to accept this work and I explain why.

The reviews and the ongoing discussion during the rebuttal showed that the work is interesting
with its main strength being the novel exploration of GNNs application on irregularly samples multivariate
time series.

There were many concerns raised by a reviewer regarding important  theoretical and methodological issues in the paper.
During the rebuttal phase, the authors clarified and resolved the majority of the concerns and there was an ongoing discussion among the two sides, authors and reviewer (which I have to admit was a pleasure to watch researchers communicating).
The authors took into account the feedback and revised the manuscript accordingly. Having read the edits myself, I believe the submission is substantially improved and addressed the concerns sufficiently.

I expect that this work will stimulate further research in the community and I would like to accept this.